# Neural Combinatorial Optimization for Time-Dependent Traveling Salesman Problem

**Ruixiao Yang**
Massachusetts Institute of Technology
`ruixiao@mit.edu`

**Chuchu Fan**
Massachusetts Institute of Technology
`chuchu@mit.edu`

## Abstract

The Time-Dependent Traveling Salesman Problem (TDTSP) extends the classical TSP by allowing dynamic edge weights that vary with departure time, reflecting real-world scenarios such as transportation networks, where travel times fluctuate due to congestion patterns. TDTSP violates symmetry, triangle inequality, and cyclic invariance properties of classical TSP, creating unique computational challenges. In this paper, we propose a neural model that extends MatNet from static asymmetric TSP to time-dependent settings by using an adjacency tensor to capture temporal variations, followed by a time-aware decoder. Our architecture addresses the unique challenge of asymmetry and triangle inequality violations that change dynamically over time. Beyond architectural innovations, our research reveals a critical evaluation insight: many practical TDTSP instances maintain the same optimal solution regardless of time-dependent edge weights. This exposes a fundamental limitation in current evaluation practices for TDTSP that rely solely on average travel time metrics across all instances. Such metrics fail to effectively distinguish between methods that genuinely capture temporal dynamics and those that merely perform well on static routing problems. Instead, we present extensive experiments on real-world datasets, evaluating our approach on both entire datasets and specifically filtered instances where temporal dependencies alter the optimal solution. Results show that our method achieves state-of-the-art average optimality gap on full instances and significant travel-time reduction on instances where time-aware routing saves time. These results demonstrate state-of-the-art ability to identify and exploit temporal dependencies, setting new standards for evaluating time-dependent routing problems.

## 1 Introduction

The Traveling Salesman Problem (TSP) is a widely studied optimization problem with applications in logistics and transportation. However, the classic TSP assumes static edge weights, failing to capture real-world dynamics in which travel times vary with departure time due to traffic patterns. This limitation is particularly relevant in urban environments where optimal routes during off-peak hours may become inefficient during rush hour. As cities continue to grow and delivery demands increase, the ability to optimize routes with time-dependent considerations becomes increasingly valuable for logistics operations, environmental impact reduction, and customer service.

The Time-Dependent Traveling Salesman Problem (TDTSP) [23] extends the classical TSP by incorporating time-varying edge weights. In TDTSP, the cost of traversing an edge depends not only on the distance but also on the departure time from the origin node. This time dependency reflects real-world scenarios where traffic patterns, weather conditions, or operational schedules create dynamic travel costs throughout the day. Compared with the metric TSP, TDTSP violates the symmetry, triangle inequality, and cyclic invariance properties, making it more challenging.

39th Conference on Neural Information Processing Systems (NeurIPS 2025).

While traditional exact and heuristic methods exist, finding near-optimal solutions efficiently remains difficult. Recent Neural Combinatorial Optimization approaches using deep reinforcement learning (DRL) have shown promise. However, current DRL approaches exhibit limitations in learning spatiotemporal dynamics and in their evaluation methodology. Existing models either separate the time-dependent adjacent tensor by node [38, 16] or by time [8], failing to capture both spatial and temporal structures simultaneously. Meanwhile, existing works evaluate methods across all instances in datasets, using only the average tour duration as the evaluation metric. Through our analysis of real-world datasets from 12 cities, we identify a shared instance distribution, where most instances randomly generated from a practical dataset maintain the optimal solution regardless of time-dependent edge weights. Under such a distribution, a well-designed ATSP solver could also achieve low average durations, rendering the metric insufficient to demonstrate the model's effectiveness in learning time dependencies. To overcome these limitations, we introduce a novel neural model that learns spatial and temporal structures simultaneously, and a post-processing step to improve solution quality based on the data distribution.

In summary, the contribution of the paper can be highlighted as:

1. Empirical analysis of practical TDTSP data, identifying the limitations of the evaluation method in existing DRL work and proposing a new evaluation method.

2. An end-to-end neural network model that directly encodes the time-dependent adjacency tensors, effectively capturing the complicated spatiotemporal dynamics in TDTSP.

3. An effective inference process to enhance the solution quality based on the data distribution.

4. We conduct comprehensive experiments on real-world datasets using the proposed evaluation method, demonstrating our method's state-of-the-art performance and strong support for learning spatiotemporal dependencies.

## 2    Related Work

### 2.1    Time Dependent Traveling Salesman Problem

The Time-Dependent Traveling Salesman Problem (TDTSP) is used to refer to two different problems: the scheduling problem where job time depends on the order in the sequence [30, 15, 1], and the routing problem where the traveling time depends on the distance and departure time [23, 24, 21]. In this paper, we focus on the latter one. This problem is sometimes studied within the broader framework of Dynamic TSP (DTSP) and Dynamic VRP (DVRP) [12, 38, 9, 32, 29].

Solution methods fall into three categories, each with distinct advantages and limitations:

Exact methods include Branch-and-Cut [10], Branch-and-Bound [2], and Constraint Programming [26]. While these guarantee optimal solutions, they scale poorly to practical instances, with computational complexity growing exponentially with problem size.

Heuristic approaches such as Ant Colony Optimization [12, 25, 27], Monte-Carlo [4], Neighborhood Search [36, 33], Tabu Search [17, 14], and Simulated Annealing [31] offer better scalability and can handle larger instances. However, they can still not provide a satisfactory solution within a short computational time.

Neural methods [38, 16, 8] use the Attention Model [18] structure and the REINFORCE [35] training algorithm, providing remarkably short inference time and a small optimality gap. Taking the time-dependent adjacency matrix as input, Guo et al. [16] and Zhang et al. [38] encode each row separately as a node feature, focusing on temporal structure while overlooking asymmetric spatial relations. On the contrary, Chen et al. [8] processes the adjacency matrix only at a specific time point during decoding, thereby failing to incorporate temporal structure. Our method directly encodes the time-dependent adjacency matrix, capturing the spatiotemporal dynamics simultaneously.

### 2.2    Neural Combinatorial Optimization for Routing

The pioneering work of Vinyals et al. [34] introduced Pointer Networks, a sequence-to-sequence learning method with attention to output permutations for TSP, which is effective but requires expensive labeled training data. The follow-up work [3] addresses the issue by combining pointer

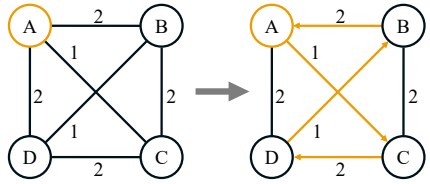
(a) TSP instance and solution

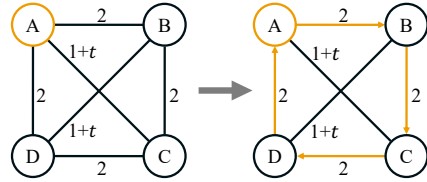
(b) TDTSP instance and solution

Figure 1: Example showing the changing weights results in a changing optimal solution. We mark the starting node and the solution in yellow. In the TSP instance, the shortest tour is ACDBA, with a traveling time of 6. The TDTSP instance, sharing the same adjacent matrix with the TSP instance at starting time $t = 0$, has the optimal solution ABCDA, which differs from the TSP instance with a traveling time of 8. Note that the tour ACDBA now has the traveling time $(1+0)+2+(1+3)+2 = 9$.

networks with reinforcement learning (RL) to train without labels. Kool et al. [18] further advanced this direction with the Attention Model (AM), which uses a multi-head attention encoder and a single-head attention decoder to construct TSP tours auto-regressively. Following the same structure, Bresson and Laurent [7] replaces the attention encoder with a transformer encoder for better performance.

The early works focus on metric TSP [34, 3, 11, 18, 28, 19, 7, 22], which can be easily encoded by node coordinates in $\mathbb{R}^{n \times 2}$. Recent advances have begun to explore the more general routing problem in non-Euclidean spaces using edge features. Kwon et al. [20] proposes the MatNet to encode the adjacency matrix for Asymmetric TSP (ATSP) with a separation of nodes' departure and arrival roles, which is also used by Gaile et al. [13]. Zhang et al. [37] proposes a variant of the graph attention network to take edge features as input. Our work follows MatNet for encoding tensor inputs.

## 3 Preliminary

In this section, we will first give a formal problem statement of TDTSP with a theorem of its hardness. Then we will briefly introduce the existing method that encodes matrix input for Asymmetric TSP.

### 3.1 Problem Statement

The Time-Dependent Traveling Salesman Problem (TDTSP) is a generalization of the classical TSP where the travel time between locations varies as a function of time. Formally, we define it as follows:

Given a complete directed graph $G = (V, E)$ where $V = \{v_1, v_2, ..., v_n\}$ is the set of nodes (cities or locations) and $E$ is the set of edges connecting these nodes. Unlike the classical TSP, where each edge $(v_i, v_j)$ has a constant cost $c_{ij}$, in TDTSP, the cost of traversing an edge depends on the departure time from the origin node.

Let $c_{ij}(t)$ denote the time needed for traveling from node $v_i$ to node $v_j$ when departing node $i$ at time $t$. We do not assume symmetry, which means $c_{ij}(t) \neq c_{ji}(t)$ in general. We assume $c_{ij}(t)$ is a continuous function for $t \in [0, T]$. The arrival time at node $v_j$ when departing from $v_i$ at time $t$ is $t + c_{ij}(t)$. We assume the salesman is not waiting at any nodes, which is valid when the property of the First-In-First-Out (FIFO) [17] holds. Then the departure time is the same as the arrival time.

The objective of TD-TSP is to find a permutation $\pi = (\pi_1, \cdots, \pi_n)$ of nodes in $V$ where $\pi_1 = v_1$ is typically designated as the depot, such that the total traveling time is minimized:

$$\min_{\pi} \sum_{k=1}^{n-1} c_{\pi_k \pi_{k+1}}(t_{\pi_k}) + c_{\pi_n \pi_1}(t_{\pi_n}), \tag{1}$$

where $t_{\pi_1} \geq 0$ is the initial departure time and $t_{\pi_k}$ is the arrival time at $\pi_k$ which is defined as

$$t_{\pi_k} = t_{\pi_{k-1}} + c_{\pi_{k-1} \pi_k}(t_{\pi_{k-1}}). \tag{2}$$

We show an example in Fig. 1. Next, we present the hardness of approximating TDTSP.

**Theorem 1** (Hardness of TDTSP). *TDTSP cannot be approximated by any $a(n)$-approximation algorithm unless P=NP, where $a(n)$ is a function that can be computed in polynomial time.*

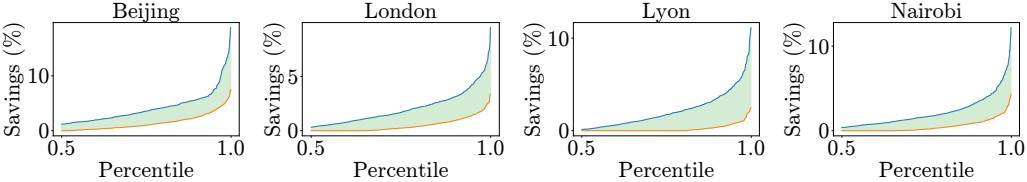

Figure 2: Distribution of travel time savings achieved by time-aware routing compared to static routing on randomly sampled instances from real-world datasets. The x-axis shows percentiles of instances, and the y-axis shows the corresponding travel time saved (in percentage). Note the long-tail distribution, indicating that significant time savings occur in a small but important subset of instances.

The proof can be found in Appendix A due to space limitations.

TDTSP can be described by the time-dependent adjacent matrix $A(t) = [c_{ij}(t)]_{i,j \in V}$. Since the continuous function $c(t)$ is usually not analytical in practice, we approximate it by interpolating samples of $\{c_{ij}(0), c_{ij}(1), \cdots, c_{ij}(T-1)\}$. In this paper, the input of TDTSP is a tensor $A[t] = \{A(t) : t = 0, 1, \cdots, T-1\}$ with linear interpolation to approximate $A(t)$.

### 3.2 Encoding Adjacent Matrix Input

To solve the asymmetric TSP (ATSP) with neural methods, Kwon et al. [20] proposes a dual graph attention layer. The static adjacency matrix $A$ of a set of node $V = \{v_1, \cdots, v_n\}$ is treated as the weights of a bipartite graph $(V^+, V^-)$, where $V^+ = \{v_1^+, \cdots, v_n^+\}$ is the set of nodes with outgoing edges and $V^- = \{v_1^-, \cdots, v_n^-\}$ is the set of nodes with incoming edges. The edge weight is defined as $d(v_i^+, v_j^-) = d(v_i, v_j) = A_{ij}$. This bipartite construction elegantly addresses the asymmetric nature of ATSP, where generally $A_{ij} \neq A_{ji}$.

The nodes in $V^+$ and $V^-$ are encoded into two separate set of vectors: $\{h_{v_i}^+\}$ and $\{h_{v_i}^-\}$. The embedding $\{h_{v_i}^+\}$ is initialized as zero vectors and $\{h_{v_i}^-\}$ as one-hot vectors. To encode the asymmetric distance information, $\{h_{v_i}^+\}$ and $\{h_{v_i}^-\}$ are iteratively updated using both current node embeddings and the edge weights $A$.

Inside each update iteration, $\{h_{v_i}^+\}$ and $\{h_{v_i}^-\}$ are processed through a dual graph attention layer, which consists of two attention-based update functions $F^+$ and $F^-$ with identical structure. Function $F^+$ updates the embeddings $\{h_{v_i}^+\}$ based on the embeddings of all nodes in $V^-$, while $F^-$ updates $\{h_{v_i}^-\}$ based on the embeddings of all nodes in $V^+$:

$$
\begin{aligned}
(h_{v_i}^+)' &= F^+(h_{v_i}^+, \{(h_{v_j}^-, A_{ij}) : v_j^- \in V^-\}), \\
(h_{v_j}^-)' &= F^-(h_{v_j}^-, \{(h_{v_i}^+, A_{ji}) : v_i^+ \in V^+\}).
\end{aligned}
\tag{3}
$$

Through this iterative process, each node accumulates information about its distance relationships with other nodes, enabling the model to learn effective representations for the ATSP.

## 4 Data Analysis

We analyze time-dependent travel time data from 12 cities across three benchmarks [26, 38, 5]. Fig. 2 shows 4 examples (remaining in Appendix C). The travel time saved by considering time-dependent edge weights follows a Pareto distribution consistently across all cities. Approximately half of the instances show no change in optimal tours despite varying edge weights, while roughly 20% of instances contribute more than 80% of total savings, resulting in low average improvement.

This distribution pattern creates two significant challenges. First, evaluating algorithms by average performance across all instances, as done in previous works [38, 16, 8], fails to distinguish between general routing capability and the specific ability to exploit time-dependent patterns. Second, the limited number of meaningful time-dependent instances complicates the learning process. For supervised learning, obtaining labels is prohibitively expensive due to the NP-hard nature of TDTSP. For DRL, quantifying the potential improvement without ground truth becomes difficult.

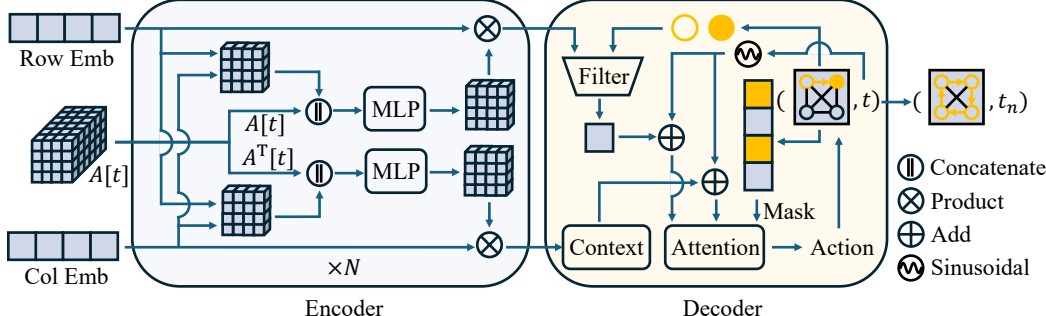

Figure 3: Architecture of our time-dependent method. The encoder processes the adjacency tensor $A[t]$ and its transpose $A^\top[t]$ (where rows and columns represent departure and arrival nodes, respectively) through $N$ iterative dual graph attention layers to produce node embeddings. The decoder then constructs the tour auto-regressively, maintaining a state of (partial tour, current time) and using masked attention with temporal embeddings to select each subsequent node.

Despite these challenges, the problem remains economically significant. In logistics markets worth trillions, even a 1% improvement yields substantial cost savings.

# 5 Methodology

To extend current methods to TDTSP and develop an effective solver in practical scenarios, we must address two key challenges:

- The complex spatio-temporal features in TDTSP, where the adjacency matrix is expanded to $A[t]$ include one more dimension of time $t$, making neural methods challenging to train.
- The Pareto distribution of TDTSP solutions, where problem instances whose optimal solutions differ from their static counterparts are rare but critically important in practice.

Our methodology addresses these challenges through two components: (1) a neural architecture directly encoding the time-dependent adjacency tensor to capture spatio-temporal structures and decode complete trajectory solutions; and (2) a post-processing refinement method enhancing solution quality while preserving computational efficiency. The following sections detail each component and demonstrate how each component addresses the identified TDTSP challenges.

## 5.1 Model Structure

Our model, shown in Fig. 3, uses an encoder-decoder architecture based on AM [18]. It takes the time-dependent adjacency tensor $A[t]$ as input, encoding it into two sets of node embeddings $\{h_{v_i}^+\}$ and $\{h_{v_i}^-\}$, representing nodes with outgoing and incoming edges, respectively. Using these embeddings, the decoder then auto-regressively constructs the full trajectory step by step.

### 5.1.1 Encoder

The goal of the encoder is to transform the adjacency tensor $A[t]$ into two sets of node embeddings $\{h_{v_i}^+\}$ and $\{h_{v_i}^-\}$. As introduced in Section 3.2, encoding a static adjacency matrix $A$ for spatial information has been well-established. The challenge lies in extending the approach to the time-dependent adjacency tensor $A[t]$ to capture both spatial and temporal information simultaneously.

For the higher dimensional $A[t]$, similar to the operation of static $A$, we treat the input time-dependent adjacency tensor $A[t]$ as a bipartite vector-weighted graph $G = (V^+, V^-, A[t])$, where $V^+$ and $V^-$ correspond to the nodes of outgoing edges and incoming edges. Each edge $(v_i^+, v_j^-)$ has a vector weight $A_{ij}[t]$ with costs at discrete time steps. The node embedding $h_v^+$ and $h_v^-$ are similarly initialized as zero and one-hot vectors and updated according to the mechanism in Eq. (3).

To facilitate the exchange of temporal information while keeping computational cost manageable, the update function $F^+$ is designed as a three-stage process. First, we compute attention scores between

outgoing and incoming nodes:

$$W^+ = \text{Score}(\{h_v^+\}, \{h_v^-\}) \tag{4}$$

where $W^+$ is an $n \times n$ matrix representing the importance of each outgoing-incoming node pair. Intuitively, this matrix represents how much attention each source node should pay to each destination node, independent of the specific time of travel.

Second, we then integrate the temporal information by concatenating these attention weights $W^+$ with the time-dependent adjacency tensor $A[t]$ along the time dimension. This combined representation is then processed through a multi-layer perceptron (MLP) that learns to compress the temporal patterns into the attention weights:

$$(W^+)' = \text{MLP}([A[t], W^+]) \tag{5}$$

This step allows the model to selectively focus on the most relevant temporal patterns for each node pair $(v_i^+, v_j^-)$ while reducing the computation from naively processing the full temporal dimension.

Finally, we update the embedding by a weighted aggregation

$$(h_v^+)' = \text{Softmax}((W_v^+)')^\top [h_{v_1}^+, \cdots, h_{v_n}^+]. \tag{6}$$

The softmax operation ensures that the attention weights are normalized, allowing the model to focus on the most relevant connections while maintaining a consistent scale for the embeddings.

The update function $F^-$ shares the same update mechanism for nodes with incoming edges. We stack $N$ of these dual graph attention layers, allowing the embeddings to progressively refine their representation of spatio-temporal patterns. The final embeddings $\{h_{v_i}^+\}^{(N)}$ and $\{h_{v_i}^-\}^{(N)}$ capture the rich spatio-temporal structure of the problem instance and are passed to the decoder.

### 5.1.2   Decoder

Our decoder constructs the complete tour in an auto-regressive manner, meaning it sequentially predicts the next node $\pi_{l+1}$ at each time step based on previously selected partial tour $\pi_{1:l}$ until the entire trajectory is constructed. This approach is particularly well-suited to TDTSP, as it naturally accommodates the problem's time-dependent nature by updating temporal information at each step.

At each step of decoding, the decoder takes two state variables: the partial tour $\pi_{1:l}$ (containing $l$ visited nodes) and the current time $t_l$ that tracks when we would arrive at node $\pi_l$. The probability distribution of the next node is computed via a masked attention mechanism, where the query vector $\mathbf{q} = \text{MLP}([h_{v_{\pi_0}}^+, h_{v_{\pi_{-1}}}^+])$ is extracted from current node and the start (which is also the destination) node. The key, value, and logit key $\mathbf{k}, \mathbf{v}, \bar{\mathbf{k}} = \text{MLP}(\{h_v^-\})$ are extracted from the node set $V^-$.

To take into account the time information, we encode the current time with Sinusoidal embedding into $\mathbf{t}$, and compute the probability of the next node as

$$p_\theta(v|\pi_{1:l}) = \text{Softmax}\left[\left(\frac{(\mathbf{q}+\mathbf{t})(\mathbf{k}+\mathbf{t})^\top}{\sqrt{d_\mathbf{k}}}\right)\mathbf{v}\bar{\mathbf{k}}^\top\right]_v. \tag{7}$$

One key insight about the sinusoidal embedding's role in the attention mechanism involves its addition to node embeddings. This operation introduces temporal information while leveraging the fixed norm property of sinusoidal embeddings. Specifically:

$$(\mathbf{q}+\mathbf{t})(\mathbf{k}+\mathbf{t})^\top = \underbrace{\mathbf{q}\mathbf{k}^\top}_{node-node} + \underbrace{(\mathbf{q}+\mathbf{k})\mathbf{t}^\top}_{node-time} + \underbrace{\|t\|_2^2}_{constant!}. \tag{8}$$

### 5.2   Training

Unlike static ATSP models, our time-dependent approach cannot use the POMO [19] training framework directly. The challenge arises from the addition of edge weights, which result in asymmetric travel times for a cycle with different starting nodes. As a result, we choose to train our model with the classic REINFORCE algorithm [35], employing a rollout baseline with a reward signal equal to the negative tour completion time.

Table 1: Summary of test datasets.

| City | Size | Start Time | End Time | Interval | Sample | Dilatation (%) | Instance Size |
|------|------|-----------|----------|----------|--------|----------------|---------------|
| Beijing | 100 | 0:00 | 0:00 (+1) | 2 h | Uniform | 0 | 10, 20, 50 |
| Lyon | 255 | 6:00 | 12:30 | 6 min | Uniform | 20 | 10, 20 |
| Nairobi | 2000 | 15:00 | 21:00 | 10 min | Congestion | 0 | 10, 20 |
| London | 2000 | 15:00 | 21:00 | 10 min | Congestion | 0 | 10, 20 |

## 5.3 Inference

During inference, we employ two types of post-processing refinement methods sequentially to improve solution quality without introducing significant computational overhead.

**Mixture of Experts (MoE).** Due to the Pareto distribution property of TDTSP solutions, a large fraction of instances share the same optimal solutions with their corresponding ATSP counterparts, where a dedicated ATSP solver may excel. Leveraging this insight, we implement a Mixture of Experts (MoE) approach that evaluates solutions from both our neural model and a state-of-the-art ATSP solver, selecting the one with lower time-dependent cost.

**Local Search.** We treat the output trajectory of our method as a high-quality warm start of a near-optimal solution and apply a local search procedure to refine it further, including two-opt, three-opt, or-opt, exchange, and relocate, which are commonly used in solving TSP [6]. We limit the local search to $k = 2$ iterations to ensure computational efficiency.

This hybrid approach combines our neural model's fast inference and ability to learn spatio-temporal patterns with the strengths of existing methods, while addressing potential suboptimalities in the neural solution arising from the REINFORCE algorithm.

## 6 Experiments

In this section, we validate the effectiveness and scalability of our method on real-world datasets. Our proposed approach was programmed with PyTorch. All the experiments were conducted on a workstation with 128 Ryzen Threadripper PRO 7985WX 64-Cores CPU and 4 NVIDIA A800 GPUs.

### 6.1 Experiments Setup

**Baselines.** We compare our algorithm to typical algorithms that accept tensor input for TDTSP, which can be further categorized into three types:

1. **Exact method.** We implement a dynamic programming (DP) to solve TDTSP as the representative of the exact method. The computational cost is low on 10-node instances, expensive on 20-node instances, and unacceptable on 50-node instances.

2. **ATSP.** As shown in Sec. 4, the average travel time saving is relatively low due to the data distribution. We select the DP for ATSP and MatNet as baselines to test the ability of learning time-dependent patterns.

3. **Heuristic Algorithms**. We implement three heuristic algorithms, including the greedy algorithm (GR), the Simulated Annealing algorithm (SA), and the Ant Colony Optimization algorithm (ACO). The greedy algorithm always selects the next unvisited node with the shortest time needed. The SA follows the pseudo code provided by Zhang et al. [38] with the same parameters. Details of GR and ACO are provided in Appendix B.

To ensure rigorous experimental fairness, we implement baselines using PyTorch tensors with GPU acceleration for all methods. All evaluations use identical conditions: single GPU hardware, consistent datasets, a uniform batch size of 1024, and identical data ordering.

**Dataset.** We generate data from real-world datasets across four cities: Beijing [38], Lyon [26], Nairobi, and London [5]. These datasets contain time-dependent travel times between nodes, with

Table 2: Computational results on TDTSP.

(a) Scalability experiments.

| Method | Obj ↓ | Gap(%) ↓ | Time(s) ↓ | Obj ↓ | Gap(%) ↓ | Time(s) ↓ | Obj ↓ | Gap(%) ↓ | Time(s) ↓ |
|---|---|---|---|---|---|---|---|---|---|
| | Beijing-10 | | | Beijing-20 | | | Beijing-50 | | |
| DP, TDTSP | **2.59** | **0.00** | 15.00 | **4.27** | **0.00** | - | - | - | - |
| DP, ATSP | 2.70 | 4.43 | 5.08 | 4.46 | 4.52 | - | - | - | - |
| MatNet, ATSP | 2.71 | 4.66 | **1.06** | 4.48 | 5.18 | **2.33** | 8.38 | - | **8.61** |
| GR | 3.34 | 19.72 | **0.16** | 5.90 | 38.58 | **0.98** | 11.66 | - | 14.16 |
| SA | **2.60** | **0.06** | 226.03 | **4.30** | **0.85** | 3491.62 | 8.17 | - | 38112.78 |
| ACO | 2.95 | 14.23 | 77.86 | 4.93 | 15.51 | 314.47 | 9.40 | - | 1128.55 |
| Ours (Raw) | 2.63 | 1.60 | **1.20** | 4.39 | 3.14 | **2.66** | 8.24 | - | **9.37** |
| Ours (Imp) | **2.59** | **0.21** | 14.06 | **4.30** | **0.89** | 108.25 | **8.09** | - | 1549.88 |

(b) TDTSP results on 10 cities.

| Method | Obj ↓ | Gap(%) ↓ | Time(s) ↓ | Obj ↓ | Gap(%) ↓ | Time(s) ↓ | Obj ↓ | Gap(%) ↓ | Time(s) ↓ |
|---|---|---|---|---|---|---|---|---|---|
| | Lyon-10 | | | Nairobi-10 | | | London-10 | | |
| DP, TDTSP | **18.79** | **0.00** | 27.48 | **18.18** | **0.00** | 30.31 | **24.54** | **0.00** | 27.38 |
| DP, ATSP | 18.92 | 0.66 | 10.63 | 18.43 | 1.32 | 10.57 | 24.60 | 0.25 | 10.50 |
| MatNet, ATSP | 18.94 | 0.78 | **0.84** | 18.45 | 1.41 | **0.91** | 24.65 | 0.47 | **0.98** |
| GR | 28.09 | 50.31 | **0.45** | 25.46 | 40.29 | **0.62** | 33.38 | 36.22 | **0.70** |
| SA | **18.80** | **0.07** | 353.17 | **18.19** | **0.02** | 345.15 | **24.55** | **0.01** | 376.58 |
| ACO | 19.64 | 4.78 | 141.84 | 18.63 | 2.49 | 171.71 | 24.91 | 1.50 | 139.77 |
| Ours (Raw) | 18.95 | 0.87 | **0.78** | 18.23 | 0.28 | **0.77** | 24.64 | 0.43 | **0.99** |
| Ours (Imp) | **18.81** | **0.11** | 29.63 | **18.19** | **0.04** | 23.06 | **24.55** | **0.04** | 40.00 |

(c) TDTSP results on 20 cities.

| Method | Obj ↓ | Gap(%) ↓ | Time(s) ↓ | Obj ↓ | Gap(%) ↓ | Time(s) ↓ | Obj ↓ | Gap(%) ↓ | Time(s) ↓ |
|---|---|---|---|---|---|---|---|---|---|
| | Lyon-20 | | | Nairobi-20 | | | London-20 | | |
| DP, TDTSP | **27.64** | **0.00** | - | **24.78** | **0.00** | - | **32.29** | **0.00** | - |
| DP, ATSP | **27.76** | **0.48** | - | 25.19 | 1.67 | - | 32.49 | 0.61 | - |
| MatNet, ATSP | 27.97 | 1.19 | **2.35** | 25.29 | 2.06 | **1.95** | 32.65 | 1.11 | **2.06** |
| GR | 40.85 | 87.52 | 4.50 | 41.07 | 65.84 | 4.11 | 61.26 | 89.63 | 4.90 |
| SA | 27.83 | 0.72 | 4362.10 | **24.87** | **0.35** | 4870.34 | **32.35** | **0.19** | 5008.44 |
| ACO | 29.33 | 6.60 | 500.63 | 26.09 | 5.41 | 566.32 | 33.38 | 3.38 | 570.10 |
| Ours (Raw) | 28.13 | 1.83 | **2.63** | 25.31 | 2.85 | **2.17** | 32.38 | 0.30 | **2.79** |
| Ours (Imp) | **27.74** | **0.40** | 221.59 | **24.88** | **0.40** | 290.73 | **32.31** | **0.08** | 297.61 |

varying sampling intervals and time horizons. For Nairobi and London, we create smaller datasets by selecting the 50 most congested nodes based on the average edge-weight variance over time. From each city's data, we generate problem instances with $n \in \{10, 20\}$ nodes via downsampling (50-node instances were generated only for Beijing due to time-horizon constraints in the other cities). For each problem size and city combination, we generated 10,000 test instances. Dataset characteristics, including sampling frequencies and time periods, are summarized in Table 1.

**Hyper Parameters.** We train our model with a batch size of 1024 for instances of size 10 and 20, and 256 for instances of size 50. We use Adam optimizer with a learning rate $1 \times 10^{-4}$. We train 200, 300, and 500 epochs, respectively, with each epoch 1,280,000 data. We train MatNet with the same batch size and epochs, and a learning rate of $4 \times 10^{-4}$. All training is distributed on 4 GPUs.

Table 3: Average gaps tested on two sets of selected instances. Gaps report in %.

| | $= 0$ | $> 3\%$ | $= 0$ | $> 3\%$ | $= 0$ | $> 3\%$ | $= 0$ | $> 3\%$ |
|---|---|---|---|---|---|---|---|---|
| Model | Beijing-10 | | Lyon-10 | | Nairobi-10 | | London-10 | |
| DP | 0.00 | 9.46 | 0.00 | 4.68 | 0.00 | 4.76 | 0.00 | 3.20 |
| MatNet | 0.70 | 9.11 | 0.24 | 4.04 | 0.25 | 4.41 | 0.33 | 1.52 |
| Ours (Raw) | 1.42 | 1.70 | 0.72 | 1.25 | 0.24 | 0.22 | 0.40 | 1.00 |
| Ours (Imp) | 0.05 | 0.32 | 0.03 | 0.29 | 0.01 | 0.06 | 0.02 | 0.00 |
| Model | Beijing-20 | | Lyon-20 | | Nairobi-20 | | London-20 | |
| DP | 0.00 | 6.94 | 0.00 | 3.92 | 0.00 | 4.61 | 0.00 | 3.31 |
| MatNet | 2.31 | 6.94 | 1.03 | 2.14 | 0.86 | 4.22 | 0.89 | 2.21 |
| Ours (Raw) | 2.75 | 3.32 | 1.75 | 2.03 | 3.14 | 3.66 | 0.33 | 0.10 |
| Ours (Imp) | 0.34 | 1.34 | 0.35 | 0.69 | 0.19 | 0.88 | 0.05 | 0.10 |

## 6.2 Comparison Results

In the comparison experiments, we test both the direct output of our neural model (Raw) and the improved solution with MoE and local search (Imp). We evaluate the methods based on average tour duration, average gap to the optimal solution, and running time.

**Full Instances.** We first evaluate our method on all instances and show the results in Table 2. Our method achieves competitive performance with the best heuristic, SA, on small cases with $n = 10$. As the problem size $n$ increases, our method outperforms all baseline methods, demonstrating the effectiveness of our pipeline for TDTSP. On the Beijing dataset, our method successfully scales to 50-node instances, as shown in Table 2a, demonstrating its scalability. The strong performance across four datasets also supports the general applicability of our method. We also compare two variants of our method. The sole neural model produces a slightly worse solution but runs much faster, revealing the great potential of solving TDTSP with a learning-based method alone.

**Selected Instances.** To better understand our model's performance, we evaluate it on two distinct subsets of the data. The first contains instances where time-dependent routing offers no duration reduction, while the second includes instances where it provides a reduction greater than 3%. This threshold is based on the 90th percentile of improvement in our dataset, adapting the methodology of Melgarejo et al. [26] to our stronger static baseline. As shown in Table 3, our neural model is generally worse than MatNet in the first type of instances, but consistently outperforms both ATSP solvers. The results suggest that our method excels at capturing temporal structure but slightly falls short of MatNet in capturing spatial structure, which motivates the use of MoE during inference.

## 6.3 Ablation Study

We conduct an ablation study to evaluate the impact of key design choices and hyperparameters, including the inference improvement methods, the number of encoder layers, the learning rate, and the temporal embedding technique.

**Inference.** We assess our solution improvement methods on 10-node instances from the Beijing dataset. As shown in Table 4, both MoE and local search enhance solution quality. The MoE effectively addresses the limitations of our neural model in learning spatial structure. The local search delivers significant improvements in the first iteration, with diminishing returns thereafter. The first iteration alone yields a substantial improvement, reducing the optimality gap from 1.60% to 0.39%. A second iteration yields a more moderate gain of 0.30%, while subsequent iterations, up to the tenth, offer minimal further improvement, reaching 0.20% at a similar computational cost per iteration. Given the computational cost per iteration, we recommend using the MatNet-based MoE with a single local search iteration for large-scale problems.

**Number of layers.** As detailed in Table 5a, both training time and GPU memory consumption scale linearly with the number of encoder layers, $N$. Model performance improves substantially as $N$ increases to 3, but shows little to no further improvement for $N > 3$.

Table 4: Comparison of different improving methods on Beijing-10.

| Iterations | 0 | | 1 | | 2 | | 10 | |
|---|---|---|---|---|---|---|---|---|
| Expert | Obj | Gap(%) | Obj | Gap(%) | Obj | Gap(%) | Obj | Gap(%) |
| None | 2.6298 | 1.6017 | 2.5990 | 0.3851 | 2.5970 | 0.3034 | 2.5945 | 0.2081 |
| DP, ATSP | 2.6120 | 0.8648 | 2.5956 | 0.2472 | 2.5946 | 0.2073 | 2.5930 | 0.1467 |
| MatNet, ATSP | 2.6129 | 0.9056 | 2.5960 | 0.2640 | 2.5946 | 0.2091 | 2.5930 | 0.1470 |

Table 5: Ablation Study for structural and training parameters.

(a) Number of encoder layers $N$.

| $N$ | 1 | 2 | 3 | 4 | 5 |
|---|---|---|---|---|---|
| GPU Memory (GiB) | 2380 | 3018 | 3764 | 4478 | 5212 |
| Training Time (s / epoch) | 33.06 | 52.56 | 72.24 | 91.63 | 111.59 |
| Obj $\downarrow$ | 2.93 | 2.66 | 2.63 | 2.63 | 2.63 |
| Gap(%) $\downarrow$ | 13.43 | 2.63 | 1.80 | 1.63 | 1.60 |

(b) Learning rate.

| lr | 3e-3 | 1e-3 | 3e-4 | 1e-4 | 3e-5 | 1e-5 | 3e-6 |
|---|---|---|---|---|---|---|---|
| Obj | 4.2230 | 2.6409 | 2.6305 | 2.6293 | 2.6468 | 2.6852 | 2.8778 |
| Gap(%) | 65.7460 | 2.0525 | 1.6316 | 1.5793 | 2.2775 | 3.8439 | 11.4468 |

(c) Time embedding.

| Embedding | Sinusoidal | MLP | Linear |
|---|---|---|---|
| Obj | 2.6298 | 2.6409 | 2.6447 |
| Gap(%) | 1.6018 | 2.0443 | 2.1953 |

**Learning rate.** The results in Table 5b indicate that the optimal learning rate lies between $3e-5$ and $1e-3$. An excessively large rate ($3e-3$) causes the training to diverge, whereas a rate that is too small ($3e-6$) results in the model converging to a poor local minimum.

**Temporal embedding.** We compared three different temporal embedding methods, as shown in Table 5c. The sinusoidal embedding proved to be the most effective, outperforming both MLP-based and linear embeddings. While other embedding strategies may exist, a broader investigation is beyond the scope of this paper.

## 7   Conclusion and Discussion

In this paper, we investigated the Time Dependent Traveling Salesperson Problem (TDTSP), which extends classic TSP by incorporating dynamic edge weights to reflect real-world environmental changes. Our primary contribution is a novel neural-based approach that directly encodes time-dependent tensors to effectively capture spatial-temporal dynamics. We identified limitations in evaluating with average tour duration on full datasets—insufficient for showing effective temporal structure learning—and proposed a new evaluation method. Experimental results validate our method's state-of-the-art performance and effectiveness in capturing complex spatial-temporal structures.

Our approach shows limitations on problem instances where TDTSP solutions coincide with static ATSP counterparts, where specialized ATSP methods slightly outperform our approach. This stems from a key challenge: training with REINFORCE compromises spatial structure learning because TDTSP violates the cyclic invariance property present in ATSP. While we demonstrated mitigation through an MoE approach during inference, developing training algorithms that better preserve spatial structure learning without relying on cyclic invariance remains an important direction for future work.

## Acknowledgments

This work was partly supported by the National Science Foundation (NSF) CAREER Award #CCF-2238030 and the MITEI Future Energy Systems Center. Any opinions, findings, conclusions, or recommendations expressed in this publication are those of the authors and don't necessarily reflect the views of the sponsors.

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

Figure 4: Distribution of travel time savings achieved by time-aware routing compared to static routing on randomly sampled instances from real-world datasets. The x-axis shows percentiles of instances, and the y-axis shows the corresponding travel time saved (in percentage). Note the long-tail distribution, indicating that significant time savings occur in a small but important subset of instances.

## A   Proof of Theorem 1

**Theorem 1** (Hardness of TDTSP). *TDTSP cannot be approximated by any $a(n)$-approximation algorithm unless P=NP, where $a(n)$ is a function that can be computed in polynomial time.*

*Proof.* For any Hamiltonian problem $G = (V, E)$, let $c_{ij}(t) = 1$ for $t \leq n$ and $(i, j) \in E$ in graph. Otherwise, $c_{ij}(t) = a(n) \cdot n$. The FIFO property holds. If a Hamiltonian cycle exists, the salesman can travel along the cycle with a traveling time of $n$. Otherwise, the salesman needs at least $a(n) \cdot n$ traveling time. So $a(n)$-approximation exists if and only if P equals NP. □

## B   Baseline Algorithms

This section describes the details of the baseline algorithms we use.

### B.1   Greedy Algorithm

The pseudo code of the greedy algorithm is presented in Algorithm 1. The greedy algorithm chooses the earliest reachable node among unvisited nodes at each step based on the current node. The bottleneck of the algorithm is the computation speed of the function $c$ by interpolation.

---

**Algorithm 1** Greedy Algorithm

**Input:** node set $V$, start node $\pi_1$, start time $t_1$, time dependent cost function $c$
**Output:** A permutation $\pi$ of the node set $V$ as a TDTSP tour.
  1: **for** $i = 2, 3, \cdots, n$ **do**
  2:     $\pi_i = \arg\min_{v \in V \setminus \pi[1:i-1]} c_{\pi_{i-1}, v}(t_{\pi_{i-1}})$
  3:     $t_i = t_{i-1} + c_{\pi_{i-1}, \pi_i}(t_{\pi_{i-1}})$
  4: **end for**

---

### B.2   Ant Colony Optimization

The pseudo code of the Ant Colony Optimization is presented in Algorithm 2. For the visited node, we do not compute its score and assign a value $-\infty$.

**Algorithm 2** Ant Colony Optimization

**Input:** node set $V$, start node $\pi_1$, start time $t_1$, time dependent cost function $c$
**Parameters:** number of ants $N_{ant} = 20$, number of iterations $N_{iters} = 100$, pheromone importance $\alpha = 1$, heuristic importance $\beta = 2$, evaporation rate $\rho = 0.1$, exploitation rate $q_0 = 0.9$
**Output:** A permutation $\pi$ of the node set $V$ as a TDTSP tour.

```
 1: for u, v ∈ V do
 2:     pheromone[u][v] ← 1
 3: end for
 4: for i = 1, 2, ⋯ , N_iters do
 5:     for j = 1, 2, ⋯ , N_ant do
 6:         for k = 2, 3, ⋯ n do
 7:             for v ∈ V \ π[1 : k − 1] do
 8:                 heuristic[v] ← 1/c_{π_{k−1}v}(t_{k−1})
 9:                 score[v] ← (pheromone[π_{k−1}][v])^α · (heuristic[v])^β
10:                 q ∼ Uniform[0, 1]
11:                 if q ≤ q_0 then
12:                     π_k ← arg max_v score[v]
13:                 else
14:                     π_k ∼ softmax(score)
15:                 end if
16:                 t_k ← t_{k−1} + c_{π_{k−1}π_k}(t_{k−1})
17:             end for
18:         end for
19:         deposit ← 1/(t_n + c_{π_nπ_1}(t_n) − t_1)
20:         pheromone[π_{i−1}][π_i] ← pheromone[π_{i−1}][π_i] + deposit
21:     end for
22:     pheromone ← ρ · pheromone
23: end for
```

Table 6: Performance comparison against baselines on 100-node problem instances.

| Method | Random | Greedy | ACO | Ours |
|---|---|---|---|---|
| Obj mean (std) | 26.98 (1.60) | 8.30 (0.51) | 7.40 (0.40) | 7.43 (0.40) |
| Time (s) | 4.75 | 172.24 | 2741.25 | 42.24 |

## C  Data Analysis

We show the data analysis of the remaining eight cities in Fig. 4. The cities show similar distributions to Beijing, London, Lyon, and Nairobi.

## D  Encoder Scalability

To further evaluate the scalability of our method, we conducted an additional experiment on 100-node instances. The following results were obtained after approximately 100 hours of training for 100 epochs. As shown in Table 6, our method's solution quality (objective score of 7.43) remains comparable to the strong ACO baseline (7.40), even though our model had not yet fully converged. Critically, our method's runtime of 42 seconds is approximately 65 times faster than ACO's 2741 seconds, demonstrating that its significant computational advantage scales to larger problem sizes.

We also report the training resource used by our method, shown in Table 7. The GPU memory usage grows quadratically with the number of nodes, which is proportional to the size of the graph's adjacency matrix. The bottleneck to scale up is the training time. As the problem scales, the training time needed to converge quickly scales up and becomes difficult to track.

Table 7: Scalability analysis of computational resource

| Nodes | 10 | 20 | 50 | 100 |
|---|---|---|---|---|
| GPU Memory (GiB/batchsize) | 5212/1024 | 14326/1024 | 18748/256 | 35884/128 |
| Train Time (s/epoch) | 111.60 | 201.08 | 687.10 | 3463.77 |

Table 8: Performance using different random seeds.

| Metric | Seed | | | | | | | |
|---|---|---|---|---|---|---|---|---|
| | 0 | 132 | 1178 | 1234 | 203681 | 385172 | 759043 | 847592 |
| Obj mean | 2.6314 | 2.6299 | 2.6289 | 2.6298 | 2.6313 | 2.6300 | 2.6290 | 2.6301 |
| Obj (std) | (0.5786) | (0.5768) | (0.5765) | (0.5770) | (0.5771) | (0.5768) | (0.5768) | (0.5776) |
| Gap(%) mean | 1.6532 | 1.6065 | 1.5678 | 1.6018 | 1.6587 | 1.6097 | 1.5659 | 1.6048 |
| Gap(%) (std) | (2.4259) | (2.4204) | (2.3958) | (2.4561) | (2.4427) | (2.4195) | (2.3820) | (2.3326) |

Table 9: Cross-city generalization analysis using Beijing-10 and Lyon-10

| Method | ACO | Beijing trained only | Beijing trained, Lyon tuned 10 epochs | Lyon trained only |
|---|---|---|---|---|
| Obj | 19.64 | 19.72 | 19.09 | 18.95 |
| Gap (%) | 4.78 | 3.69 | 1.61 | 0.87 |

Table 10: Computational results with statistical significance on Beijing-10

| Method | DP, TDTSP | DP, ATSP | MatNet | GR | SA | ACO | Ours(Raw) | Ours(Imp) |
|---|---|---|---|---|---|---|---|---|
| Obj mean | 2.59 | 2.70 | 2.71 | 3.34 | 2.60 | 2.95 | 2.63 | 2.59 |
| Obj (std) | (0.57) | (0.63) | (0.63) | (0.74) | (0.57) | (0.68) | (0.58) | (0.57) |
| Gap(%) mean | - | 4.43 | 4.66 | 19.72 | 0.06 | 14.23 | 1.60 | 0.22 |
| Gap(%) (std) | - | (5.83) | (5.97) | (14.73) | (0.40) | (10.25) | (2.46) | (0.72) |

## E   Training Stability

To show the stability of training, we evaluate our trained neural policy under different random seeds. Shown in Table 8, the policies trained from different random initializations perform similarly.

## F   Cross-City Generalization

We test the cross-city generalization ability using the Beijing-10 and Lyon-10 datasets. As shown in Table 9, the performance is competitive with ACO for direct cross-city generalization, and further improved with 10 epochs of tuning on the Lyon dataset, showing superior cross-city generalization.

## G   Experiment Statistical Significance

In this section, we present the experiment results, including both the mean and the standard deviation. The Beijing-10 results demonstrate that standard deviations of the gap on the 10000 instances often match or exceed the mean of the gaps. This table confirms our finding that the time saved by considering the time-varying edge weights follows a Pareto distribution.

Table 11: Ablation Study for structural and training parameters.

(a) Number of encoder layers $N$.

| $N$ | 1 | 2 | 3 | 4 | 5 |
|---|---|---|---|---|---|
| GPU Memory (GiB) | 2380 | 3018 | 3764 | 4478 | 5212 |
| Training Time (s / epoch) | 33.06 | 52.56 | 72.24 | 91.63 | 111.59 |
| Obj $\downarrow$ | 2.93 (0.64) | 2.66 (0.58) | 2.63 (0.58) | 2.63(0.58) | 2.63 (0.58) |
| Gap(%) $\downarrow$ | 13.43 (9.02) | 2.63 (3.33) | 1.80 (2.59) | 1.63 (2.44) | 1.60 (2.45) |

(b) Learning rate.

| lr | 3e-3 | 1e-3 | 3e-4 | 1e-4 | 3e-5 | 1e-5 | 3e-6 |
|---|---|---|---|---|---|---|---|
| Obj mean | 4.2230 | 2.6409 | 2.6305 | 2.6293 | 2.6468 | 2.6852 | 2.8778 |
| Obj (std) | (0.7554) | (0.5773) | (0.5766) | (0.5775) | (0.5800) | (0.5814) | (0.6259) |
| Gap(%) mean | 65.7460 | 2.0525 | 1.6316 | 1.5793 | 2.2775 | 3.8439 | 11.4468 |
| Gap(%) (std) | (22.6098) | (2.8460) | (2.4443) | (2.4065) | (3.0714) | (4.2814) | (8.2724) |

(c) Time embedding.

| Embedding | Sinusoidal | MLP | Linear |
|---|---|---|---|
| Obj mean (std) | 2.6298(0.5770) | 2.6409 (0.5783) | 2.6447 (0.5790) |
| Gap(%) mean (std) | 1.6018 (2.4561) | 2.0443 (2.8317) | 2.1953 (2.9547) |

