# OpenReview forum: "Neural Combinatorial Optimization for Time-Dependent Traveling Salesman Problem"
_NeurIPS.cc/2025/Conference — NeurIPS 2025 poster_

### Official Review · Reviewer_n5Wo · 2025-06-21

**Clarity:** 4
**Significance:** 2
**Originality:** 2
**Rating:** 4
**Confidence:** 4

**Summary:**

The paper develops a neural combinatorial optimization method for time dependent travelling salesman problem. This extends previous works on neural combinatorial optimization methods for TSP, adding a time dependent component. The architecture extends MatNet to incorporate temporal information. The paper also shows that only a small fraction of instances from real-world distribution show substantial improvement when taking temporal information into account and propose selecting between the best solution found using only static information with that found using temporal information and performs further local search.

**Questions:**

1. For figure 2, what are the two lines shown and the region between the lines? Is that the confidence interval and if it is, is it one or two standard deviation?
2. For comparison with simulated annealing, is the SA implementation also parallelized with GPU for fair comparison?
3. For question 7 of the checklist, "Question: Does the paper report error bars suitably and correctly defined or other appropriate information about the statistical significance of the experiments?", you answered "Answer: [No]. Justification: Error bars do not support our conclusion." Is it not possible to increase the test sample size so that we can trust your experimental conclusions?

**Ethical Concerns:**

["NO or VERY MINOR ethics concerns only"]

**Final Justification:**

In the rebuttal, the authors provided sample size and variance information, which would be enough to compute error bars for their results. They also committed to adding error bars to their results in the paper, which I think would be useful. The authors also clarified that the results provided is enough to infer that their method provide good improvement over the baseline on the subset of problems where temporal information is important. After rebuttal clarification, I think that the work have made useful progress. In addition, I think that time dependent TSP is a useful direction for neural combinatorial optimization, hence I have increased my rating.

**Limitations:**

Yes.

**Paper Formatting Concerns:**

None.

**Quality:**

2

**Strengths And Weaknesses:**

Strengths:
- The work is competently done.
- The paper is clearly written.
- The analysis showing where the improvements are coming from is useful. In cases where temporal information gives more than 3% gain, the method does show useful improvements.

Weaknesses:
- Novelty is not that high. The method appears to be a straightforward adaptation of MatNet to incorporate temporal information.
- On average, results are similar to MatNet. Results are only better by selecting the better of the two methods and doing additional local search.

---

> ### Author Rebuttal · Authors · 2025-07-30
>
> We thank the reviewer for acknowledging the competent execution of our work, the clarity of our writing, and the usefulness of our analysis in identifying the sources of improvement. We hope that our responses below address the concerns raised by the reviewer.
>
> # Summary
>
> In brief, we have:
>
> 1. Clarified Figure 2 shows max/min savings (not confidence intervals) to emphasize the Pareto distribution pattern;
> 2. Confirmed fair GPU-accelerated comparison with SA using identical conditions across all baselines;
> 3. Added variance reporting (mean±std) across 10,000 instances per scenario, showing high variance aligns with the Pareto distribution;
> 4. Addressed novelty concerns by explaining non-trivial challenges: asymmetric temporal dimensions, dynamic continuous time handling, and 3D tensor complexity beyond MatNet's 2D foundation;
> 5. Clarified that the similar average performance is misleading because ~50% of instances don't benefit from temporal information, masking that our method significantly outperforms MatNet on the instances where time-dependency actually matters.
>
> # Detailed Reply
>
> ### 1. Clarification on Figure 2 visualization
>
> **Reviewer Concern:** *What do the two lines and the shaded region represent in Figure 2? Are these confidence intervals with one or two standard deviations?*
>
> **Response:** The lines represent the **maximum** and **minimum savings** across instances, not confidence intervals. This visualization emphasizes our key finding that gains from time-dependent edge weights follow a Pareto distribution—most instances show minimal improvement while a subset demonstrates substantial benefits.
>
> ### 2. Fair comparison with Simulated Annealing
>
> **Reviewer Concern:** *Is the simulated annealing implementation parallelized on a GPU for fair comparison?*
>
> **Response:** Yes, **all baselines, including SA**, are implemented with GPU acceleration for rigorous experimental fairness. **All evaluations employ identical conditions**: single GPU hardware, consistent datasets, uniform batch size (1024), and identical data ordering. Complete baseline implementations are available in our supplementary materials (folder *rl4co/rl4co/baselines*).
>
> ### 3. Statistical significance and error reporting
>
> **Reviewer Concern:** *Given your "No" answer to checklist question 7 about error bars, why not increase sample size to ensure trustworthy experimental conclusions?*
>
> **Response:** We appreciate this important point about statistical rigor. Our experimental design already uses **10,000 instances per scenario**, providing a robust statistical foundation. While variance reporting isn't standard in neural combinatorial optimization literature [8, 16, 20, 38], we recognize its value.
>
> Following your feedback, we report results in mean(std) format below. We will turn this table into error bars in the camera-ready version of the paper.​ The Beijing-10 results demonstrate that standard deviations of the gap on the 10000 instances often match or exceed the mean of the gaps.​ This table confirms our finding that the time saved by considering the time-varying edge weights follows a Pareto distribution.
>
> | Method       | Obj         | Gap           |
> | ------------ | ----------- | ------------- |
> | DP, TDTSP    | 2.59 (0.57) | \             |
> | DP, ATSP     | 2.70 (0.63) | 4.43 (5.83)   |
> | MatNet, ATSP | 2.71 (0.63) | 4.66 (5.97)   |
> | GR           | 3.34 (0.74) | 19.72 (14.73) |
> | SA           | 2.60 (0.57) | 0.06 (0.40)   |
> | ACO          | 2.95 (0.68) | 14.23 (10.25) |
> | Ours(Raw)    | 2.63 (0.58) | 1.60 (2.46)   |
> | Ours(Imp)    | 2.59 (0.57) | 0.22 (0.72)   |
>
> ### 4. Novelty Concerns
>
> **Reviewer Concern:** *The novelty appears limited—seems like a straightforward MatNet adaptation for temporal information.*
>
> **Response:** We believe this is a misunderstanding and would like to address the novelty concern.
>
> **Novel Technical Contributions:** While building on MatNet's foundation, extending to time-dependent problems required solving non-trivial challenges, not direct adaptation. Here we list the technical challenges we met and our contributions to solving them.
>
> 1. **Asymmetric Dimensional Integration**: MatNet's dual attention naturally handles symmetric spatial dimensions (outgoing/incoming nodes), but the time dimension is fundamentally asymmetric with directionality and causality that spatial dimensions lack. Our innovation​ of the vectorized dual attention mechanism integrates temporal information while respecting time's directional nature and preserving MatNet's spatial symmetry handling.
> 2. **Dynamic Continuous Time Handling**: MatNet processes static discrete indices where the (static) adjacency matrix A[i,j] remains constant throughout decoding. However, time is a continuous and evolving variable. The current time updates during decoding require real-time edge cost interpolation from discrete time samples. Our time-aware decoder uses dynamic interpolation and sinusoidal embeddings to handle continuous temporal states.
> 3. **Tensor Structure Complexity Handling**: MatNet's elegant bipartite graph representation is designed for 2D matrices, but extending to 3D tensors A[t] breaks the natural bipartite structure since edges now have vector weights instead of scalars. We extended the bipartite representation to handle vector-weighted edges with new attention mechanisms for processing temporal vectors rather than scalar ones.
>
> Besides technical novelties, we also want to emphasize our novelty in identifying the fundamental evaluation problem in TDTSP research.
>
> **Novel Methodology Contributions:** In addition to the above algorithmic contributions, we identify and solve a fundamental evaluation problem in TDTSP research that has led to misleading conclusions in prior work, thereby providing valuable insights for future work. We identified that the single average performance value is a misleading metric (see detailed response to Q5) and therefore changed the metric to the performance on selected instances where time-dependency has a significant impact, as discussed in Section 6.2, lines 276-281.
>
> ### 5. Performance comparison with MatNet
>
> **Reviewer Concern:** *Results appear similar to MatNet on average, only improving when selecting the better method and adding local search.*
>
> **Response:**
> We appreciate the reviewer for this observation. However, this is a misleading impression because the dataset comprises two distinct subsets. One consists of instances that, considering time information, give little improvement, while the other consists of instances with significant improvements. This also indicates that the *simple, single average performance* is a poor metric for evaluating the performance of different methods.
>
> **Key insights:** The **single average performance is misleading for TDTSP**. As shown in Section 4, ~50% of instances have identical optimal solutions regardless of time-dependency. This allows spatial-only methods like MatNet to achieve good averages without learning any temporal patterns.​​ This explains why prior TDTSP papers report similar performance across fundamentally different approaches. However, for the rest of the instances, MatNet performs poorly. Therefore, we argue that this is an inherited problem of the dataset. We believe that the different instances from different distribution categories should be evaluated separately, instead of using a single metric of the average performance.
>
> **In instances where temporal dependencies matter (Table 3), our method significantly outperforms MatNet.** For instances with >3% potential savings, our raw model achieves 1.70% vs 9.11% gaps (Beijing-10) and 3.32% vs 6.94% (Beijing-20), demonstrating effective temporal learning. This means that, although the average performance is similar, **the distribution of the gaps is fundamentally different** on different instances. The gap of MatNet comes from the lack of time-dependent information. In contrast, the gap of our method comes from the inadequate spatial capability.
>
> Follow this insight, **the MoE approach is principled system design**, not simple cheating. Our analysis reveals that instances naturally split into temporal-dependent vs. spatial-only categories. A hybrid approach leveraging this structure is both theoretically sound and practically valuable.
>
> In summary, the similarity in average performance is a misleading conclusion due to the poor metric used for evaluation. Our method does **capture the temporal information** well, and **better evaluation methodologies are needed** to assess temporal reasoning capabilities in neural combinatorial optimization.

---

> > ### Comment · Reviewer_n5Wo · 2025-08-04
> >
> > Thanks for the clarification. I still do not understand Figure 2. I assume that you would sort the data according to the Savings and then plot the Savings at different percentiles. But that must be wrong as there is no min/max at each percentile if done that way. How do you plot the min/max at each percentile? Also, do you have MatNet (Imp) results for Table 3 for comparison to show that the improvement is coming from your new network, and not from local search and other standard methods?

---

> > > ### Author Response · Authors · 2025-08-04
> > >
> > > Thank you for your reply. We hope the follow-up responses below address your concerns.
> > >
> > > **Regarding Figure 2:**
> > >
> > > Thank you for pointing out the confusion. Let me clarify our methodology:
> > >
> > > For each city, we generate 1000 **scenarios** (each representing a given depot and city configuration). We then vary the starting time of each scenario to create multiple **instances** (scenario + starting time). This gives us a gap array of shape [num_scenario, num_start_time] for each city.
> > >
> > > To create Figure 2: We sort instances with the **same starting time** by their optimality gap, creating one curve per starting time across the 1000 scenarios. Rather than showing all curves (which would be cluttered), we plot the envelope - the max and min values at each percentile across all starting times. This makes the figure more readable while preserving the key information about performance distribution.
> > >
> > > We will update our description to clearly distinguish between scenarios and instances for better clarity.
> > >
> > > **Regarding the improvement attribution:**
> > >
> > > Our experimental design isolates the contributions of each component through Tables 3 and 4:
> > >
> > > **Table 3** compares our model against MatNet. Our model excels at reducing optimality gaps when Static TSP has large gaps, demonstrating its ability to capture spatial-temporal relationships. However, our model performs slightly worse than MatNet when Static TSP finds optimal solutions (cases where time-dependency doesn't change the optimal solution). This is expected since we cannot use POMO due to TDTSP breaking cyclic invariance, while MatNet benefits from this technique. This matters in practice **because such cases take significant weight** due to the distribution issue we found in Section 4, which leads to our design of MoE during inference.
> > >
> > > **Table 4** validates **both** the MoE and the local search. Vertically, MoE shows improvement over the raw model. Horizontally, local search progressively reduces the optimality gap, demonstrating its effectiveness.
> > >
> > > The experimental design allows us to show each component's contribution independently without requiring MatNet (Imp).

---

> > > > ### Comment · Reviewer_n5Wo · 2025-08-04
> > > >
> > > > Thanks for the clarifications. I am slightly more positive now, going into discussion with the area chair and other reviewers.

---

> > > > > ### Author Response · Authors · 2025-08-05
> > > > >
> > > > > Thank you for your thoughtful reconsideration and continued discussion of our work. We appreciate your more positive view and remain committed to addressing any remaining concerns you may have.

---

### Official Review · Reviewer_mtMa · 2025-06-30

**Clarity:** 3
**Significance:** 2
**Originality:** 3
**Rating:** 4
**Confidence:** 4

**Summary:**

This paper addresses the Time-Dependent Traveling Salesman Problem (TDTSP), a more realistic version of the Traveling Salesman Problem where travel times on edges are not static but change based on the departure time. The authors highlight that TDTSP violates fundamental properties of classical TSP like symmetry and triangle inequality, making it computationally challenging. Existing neural combinatorial optimization methods often fail to capture both spatial and temporal dynamics simultaneously, or their evaluation methodologies are insufficient for TDTSP.

The key insight of the paper is an empirical analysis revealing that in many practical TDTSP instances, the optimal solution (the sequence of visited nodes) remains the same regardless of time-dependent edge weights. This exposes a limitation in prior evaluation methods that rely solely on average tour duration, as such metrics cannot effectively differentiate between methods truly exploiting temporal dynamics and those merely performing well on static routing.

To overcome these challenges, the paper proposes:

A novel end-to-end neural network model designed to directly encode the time-dependent adjacency tensor, thereby capturing complex spatio-temporal dynamics simultaneously. The model uses an encoder-decoder architecture with dual graph attention layers in the encoder and a time-aware decoder that incorporates sinusoidal embeddings for temporal information.

An effective inference process that enhances solution quality based on the data distribution, specifically employing a Mixture of Experts (MoE) approach (combining their neural model with a state-of-the-art Asymmetric TSP (ATSP) solver) and a local search refinement.

A new evaluation methodology that accounts for instances where temporal dependencies truly alter the optimal solution, rather than just relying on average performance across all instances.

The paper presents extensive experiments on real-world datasets from cities like Beijing, Lyon, Nairobi, and London, demonstrating state-of-the-art performance in terms of average optimality gap and significant travel time reduction on instances where time-aware routing provides actual benefits. This establishes new standards for evaluating routing problems with temporal dependencies.

**Questions:**

Question: Could the authors provide more insights into the robustness of the proposed neural model? For example, how sensitive are the results to variations in the number of dual graph attention layers (N) or different initialization strategies?
Suggestion: Conduct ablation studies or sensitivity analyses on key hyperparameters (e.g., varying the number of attention layers, exploring different learning rates for REINFORCE, or alternative temporal encoding methods beyond sinusoidal embeddings).

Question: For real-world logistics, TDTSP instances can involve hundreds or thousands of nodes. The current experiments primarily focus on N=10, 20, and 50 nodes. How do the authors envision scaling this approach to much larger instances, given the reported inference times for N=50 (e.g., ~1550s for Beijing-50)?

Question: The paper mentions using the REINFORCE algorithm for training due to the asymmetric traveling times. REINFORCE is known for high variance. Can the authors elaborate on how they managed training stability (e.g., specific baselines, large batch sizes)? Were other policy gradient methods (e.g., Actor-Critic, PPO) considered, and if so, what were the reasons for choosing REINFORCE?

**Ethical Concerns:**

["NO or VERY MINOR ethics concerns only"]

**Limitations:**

Yes.

**Paper Formatting Concerns:**

No.

**Quality:**

3

**Strengths And Weaknesses:**

Strengths:

Quality: The paper demonstrates a high level of technical quality. The proposed neural architecture for encoding time-dependent adjacency tensors is well-designed to capture spatio-temporal dynamics. The integration of sinusoidal embeddings and the inference process with Mixture of Experts (MoE) and local search show a sophisticated understanding of both neural network design and combinatorial optimization heuristics. The experimental setup is robust, comparing against various baselines (exact methods, ATSP solvers, heuristics) on real-world datasets from multiple cities, which strengthens the validity of their claims. The results indicate state-of-the-art performance, especially in instances where time-awareness is critical.

Clarity: The paper is generally well-structured and easy to follow. The problem statement for TDTSP is clearly defined. The proposed model's architecture (encoder-decoder, dual graph attention, time-aware decoder) is explained with good accompanying diagrams (Figure 3). The rationale behind each component, such as the three-stage process for the encoder's update function, is clearly articulated. The empirical analysis of TDTSP data distribution (Pareto distribution) and its implications for evaluation are presented concisely and are supported by Figure 2.

Significance: The paper makes several significant contributions.
Novel Evaluation Methodology: The identification of the "Pareto distribution" in real-world TDTSP instances and the proposal of a new evaluation method that focuses on instances where temporal dependencies truly matter is a critical and highly significant contribution. This insight can re-shape how TDTSP algorithms are assessed in future research.

Effective Model for TDTSP: The proposed neural model effectively tackles the complex spatio-temporal nature of TDTSP, which has been a challenge for existing neural methods. Its state-of-the-art performance on real-world datasets demonstrates its practical value.

Practical Relevance: TDTSP has immense practical applications in logistics and transportation, where even small improvements can lead to substantial cost savings. By providing a more effective and accurately evaluated solution, the paper has a high potential for real-world impact.

Originality: The originality of the paper lies primarily in two areas:
Empirical Data Analysis and New Evaluation: The discovery of the Pareto distribution of savings in real-world TDTSP instances and the subsequent proposal for a refined evaluation methodology is a truly original insight. This challenges existing benchmarks and provides a more meaningful way to assess performance.

Direct Encoding of Time-Dependent Tensor: While the base encoder-decoder framework and dual graph attention layers build upon prior work (e.g., MatNet, Attention Model) , the specific approach to directly encode the higher-dimensional time-dependent adjacency tensor and integrate temporal information throughout the attention mechanism is novel. The use of MoE in inference to leverage the ATSP solution for "static" instances is also a smart and original engineering contribution.


Weaknesses:

Clarity on "Online Refinement": While the paper mentions "online refinement methods", the term "online" might be misleading if it refers strictly to inference-time enhancements. It should be clarified if the MoE and local search components involve any real-time adaptation or merely post-processing of the neural model's output. The current description implies they are sequential post-processing steps.

Limited Deep Dive into Model Sensitivity: The paper does not thoroughly explore the sensitivity of the model to different hyperparameters (e.g., number of dual graph attention layers (N), learning rate) or variations in the sinusoidal embedding. Such analysis would provide more insights into the model's robustness.

Scalability for Larger Instances (my major concern): While the paper shows good scalability up to 50 nodes for Beijing, and 20 nodes for other cities, TDTSP in real-world logistics can involve hundreds or thousands of nodes. The computational time for "Ours (Imp)" on Beijing-50 (1549.88s) is quite high, suggesting that further work on scalability for very large instances (at least 100) is needed, or a discussion of trade-offs.

Training Complexity/Algorithm: The paper states that it cannot use the POMO training framework and instead relies on the REINFORCE algorithm. While REINFORCE is a classic choice, it is known for high variance. A brief discussion on how training stability or convergence was managed, or whether alternative lower-variance policy gradient methods were considered, would be beneficial.

---

> ### Author Rebuttal · Authors · 2025-07-30
>
> We thank the reviewer for recognizing the high technical quality of our neural architecture for encoding time-dependent adjacency tensors, the clarity of our problem formulation and model design, and the significance of our contributions to TDTSP research. We are particularly grateful for the acknowledgment of our novel evaluation methodology that identifies the Pareto distribution in real-world TDTSP instances—a critical insight that can reshape how TDTSP algorithms are assessed—as well as our original approach to directly encoding higher-dimensional time-dependent tensors and the practical relevance of our work for logistics and transportation applications. We hope that our responses below address the concerns raised by the reviewer.
>
> # Summary
>
> In brief, we have:
>
> 1. Conducted comprehensive ablation studies on encoder layers, learning rates, and temporal embeddings;
> 2. Demonstrated scalability with 100-node experiments, achieving comparable ACO performance but 65× faster
> 3. Addressed scalability concerns with flexible local search trade-offs and preprocessing strategies for practical applications.
> 4. Provided multi-seed stability analysis confirming REINFORCE reliability (obj: 2.630±0.01, gap: 1.61±0.05 across 8 seeds);
> 5. Clarified "online refinement" terminology - will remove "online" and specify these are post-processing steps during inference;
>
> # Detailed Reply
>
> ### 1. Model Robustness and Hyperparameter Sensitivity
>
> **Reviewer Concern:** *Could the authors provide more insights into the robustness of the proposed neural model through ablation studies on key hyperparameters?*
>
> **Response:** Beyond MoE and local search ablation (Table 4), we conducted additional component analysis examining **temporal embeddings**, **encoder layer numbers**, and **learning rate** sensitivity.
>
> | Layer N    | 1            | 2           | 3           | 4           | 5           |
> | :--------- | :----------- | :---------- | :---------- | :---------- | :---------- |
> | Train time | 33.06        | 52.66       | 72.23       | 91.63       | 111.59      |
> | GPU Mem    | 2380         | 3018        | 3764        | 4478        | 5212        |
> | Obj        | 2.93 (0.64)  | 2.66 (0.58) | 2.63 (0.58) | 2.63 (0.58) | 2.63 (0.58) |
> | Gap        | 13.43 (9.02) | 2.63 (3.33) | 1.80 (2.59) | 1.63 (2.44) | 1.60 (2.46) |
>
> **Layer Num**: The training time and the GPU memory increase linearly with the number of layers. The performance rises quickly when N<3, while it shows little to no improvement after N>=3.
>
> | lr   | 3e-3              | 1e-3            | 3e-4            | 1e-4            | 3e-5            | 1e-5            | 3e-6             |
> | :--- | :---------------- | :-------------- | :-------------- | :-------------- | :-------------- | :-------------- | :--------------- |
> | Obj  | 4.2230 (0.7554)   | 2.6409 (0.5773) | 2.6305 (0.5766) | 2.6293 (0.5775) | 2.6468 (0.5800) | 2.6852 (0.5814) | 2.8778 (0.6259)  |
> | Gap  | 65.7460 (22.6098) | 2.0525 (2.8460) | 1.6316 (2.4443) | 1.5793 (2.4065) | 2.2775 (3.0714) | 3.8439 (4.2814) | 11.4468 (8.2724) |
>
> **Learning Rate**: The best learning rate is between 3e-5 and 1e-3. A too large learning rate (3e-3) makes the model diverge, and a too small learning rate (3e-6) gets stuck at a bad local minimum.
>
> | Temporal Encoder | Sinusoidal      | MLP              | Linear          |
> | :--------------- | :-------------- | :--------------- | :-------------- |
> | Obj              | 2.6298(0.5770)  | 2.6409 (0.5783)  | 2.6447 (0.5790) |
> | Gap              | 1.6018 (2.4561) | 2.0443 (2.8317)% | 2.1953 (2.9547) |
>
> **Temporal embedding**: The sinusoidal embedding is the most helpful one, compared with the MLP embedding and the Linear embedding. There are potentially other effective embedding methods, but that is beyond the scope of the paper.
>
> These results, along with individual component impact assessments, will be detailed in the appendix.
>
> ### 2. Scalability to Large-Scale Instances
>
> **Reviewer Concern:** *How do the authors envision scaling this approach to much larger instances, given the reported inference times for N=50 (e.g., ~1550s for Beijing-50)?*
>
> **Response:** We address scalability through three complementary strategies:
>
> **Good Model Scalability**: We conducted new experiments training our method on **100-node** cases with 100 epochs (not converged yet), taking ~100 hours, showing comparable performance with ACO. (7.4327 (0.3962) vs 7.3997(0.4007)) while significantly outperforming the running speed (42.24s vs. 2741.25s). This shows that our neural network without the local search can learn larger-sized problems with acceptable learning time, while keeping a **short inference time**.
>
> | Method         | Random      | Greedy     | ACO        | Ours       |
> | :------------- | :---------- | :--------- | :--------- | :--------- |
> | Obj mean (std) | 26.98(1.60) | 8.30(0.51) | 7.40(0.40) | 7.43(0.40) |
> | Time           | 4.75        | 172.24     | 2741.25    | 42.24      |
>
> We also test the scaling of GPU memory usage and the training time required. From the table, the GPU memory grows quadratically. At the scale of 100 nodes, it requires a memory usage of 36GB and training time of an hour when setting the batch size to 128. Such training cost is **acceptable in the application**, validating the scalability of our method.
>
> | Nodes                      | 10        | 20         | 50        | 100       |
> | :------------------------- | :-------- | :--------- | :-------- | :-------- |
> | GPU Memory (GiB/batchsize) | 5212/1024 | 14326/1024 | 18748/256 | 35884/128 |
> | Train Time (s/epoch)       | 111.60    | 201.08     | 687.10    | 3463.77   |
>
> **Flexible local search trade-off**: The reported 1,550s includes extensive local search, including 2 iterations of 5 sequential operations (2-opt, 3-opt, exchange, relocate, or-opt) per iteration. There are multiple choices for reducing the running time by sacrificing the performance:
>
> - Using 1 iteration instead of 2
> - Using fewer local search operations
> - Removing local search entirely for time-critical applications
>
> **Preprocessing in practice**: For real-world scenarios like last-mile delivery, **regional customer clustering** is a common strategy that can reduce problem size without significantly compromising performance, making our approach suitable for large-scale deployment without training the full-size model.
>
> ### 3. Training Stability with REINFORCE
>
> **Reviewer Concern:** *How was training stability managed given REINFORCE's known high variance? What were the reasons for choosing REINFORCE?*
>
> **Response:** Although REINFORCE is known for high variance, **it showed quite good stability in our experiments**. The following table reports an extra experiment of training with different random seeds. The final converged states give quite stable objective: 2.630±0.01 and optimality gap: 1.61±0.05 across 8 seeds. This stability also aligns with established TDTSP literature [8,16,38], where no instability issues are reported. The observation validates our algorithmic choice of REINFORCE for TSP-related problems.
>
> | Seed | 0               | 132             | 1178            | 1234            | 203681          | 385172          | 759043          | 847592          |
> | ---- | --------------- | --------------- | --------------- | --------------- | --------------- | --------------- | --------------- | --------------- |
> | Obj  | 2.6314 (0.5786) | 2.6299 (0.5768) | 2.6289 (0.5765) | 2.6298 (0.5770) | 2.6313 (0.5771) | 2.6300 (0.5768) | 2.6290 (0.5768) | 2.6301 (0.5776) |
> | Gap  | 1.6532 (2.4259) | 1.6065 (2.4204) | 1.5678 (2.3958) | 1.6018 (2.4561) | 1.6587 (2.4427) | 1.6097 (2.4195) | 1.5659 (2.3820) | 1.6048 (2.3326) |
>
> We have also implemented the PPO before, but found it not performing better than REINFORCE, with difficulty in the critic network learning. We see very few recent works about applying PPO for learning TSP and its variants, like *Enhanced Reinforcement Learning for TSP: Proximal Policy Optimization with Graph Attention and Entropy Regularization* in 2024 and *Solving the Traveling Salesman Problem with Drones Using Proximal Policy Optimization and Deep Reinforcement Learning* in 2025. While they show that PPO can work in some cases, there is still no comprehensive study comparing different learning algorithms and providing guidelines for selecting learning algorithms for TSP variants.
>
> ### 4. Clarification on "Online Refinement"
>
> **Reviewer Concern:** *Is the "online refinement" real-time adaptation or post-processing?*
>
> **Response**: We thank the reviewer for identifying the ambiguity. Our refinement methods (MoE and local search) are **post-processing steps during inference**, not real-time adaptations. We will remove the potentially misleading term "online" and explicitly describe these as inference-time post-processing steps in our revision.

---

> > ### Comment · Reviewer_mtMa · 2025-08-04
> >
> > Thanks for your response. Experiments are clear and informative.

---

> > > ### Author Response · Authors · 2025-08-05
> > >
> > > Thank you for your positive review and constructive comments! We appreciate your acknowledgement of our rebuttal.

---

### Official Review · Reviewer_Zrkp · 2025-07-01

**Clarity:** 3
**Significance:** 2
**Originality:** 2
**Rating:** 4
**Confidence:** 4

**Summary:**

This paper proposes a neural model that encodes the time-dependent adjacency tensor to address asymmetry and triangle inequality violations. Beyond architectural innovations, the research highlights a critical evaluation insight: many practical TDTSP instances retain the same optimal solution despite time-dependent edge weights, exposing a fundamental flaw in current evaluation practices that rely solely on average travel time metrics.

**Questions:**

See strengths and weaknesses.

**Ethical Concerns:**

["NO or VERY MINOR ethics concerns only"]

**Final Justification:**

The author's response has addressed most of my concerns, and I have accordingly updated my evaluation to "borderline accept". The authors are advised to incorporate the suggested revisions in detail when preparing the final version of the manuscript.

**Limitations:**

Yes

**Quality:**

3

**Strengths And Weaknesses:**

Strengths:

The paper identifies critical limitations in the evaluation of existing algorithms and extends prior methodologies, with experimental results demonstrating notable advantages. Additionally, the logical structure of the paper is well-organized, facilitating clear comprehension of the research framework and conclusions.

Weaknesses:

1.	The paper proposes a time-dependent adjacency tensor encoding and time-aware decoder for TDTSP, but the abstract’s claim to address asymmetry and triangle inequality requires clarification. While the authors frame this as a novel contribution, Kwon et al. [20] already resolved these issues for static ATSP using matrix encoding. The extension to TDTSP is valuable, but the abstract must explicitly acknowledge the foundational work in [20] and highlight how the temporal dimension introduces unique innovations (e.g., tensor-based spatiotemporal integration). Without this distinction, the novelty of addressing asymmetry is diminished.

2.	The model’s ability to "simultaneously capture spatiotemporal structures" needs explicit comparison with prior works (e.g., [8,16,38]). Current neural approaches for TDTSP either separate temporal and spatial processing (e.g., [16] by node, [8] by time) or focus on static ATSP ([20]). To validate the claimed advantage, the authors should: 1) Provide a comparative diagram in the Introduction/Methodology showing how their tensor encoding differs from matrix-based methods in [20] and temporal segmentation in [8,16]. 2) Quantify the mechanism through which the tensor structure models time-space interactions (e.g., via attention layers that process temporal and spatial features jointly).

3.	This paper describes the three-stage process in line 181, but Fig. 3 is not clearly labeled. It is recommended to modify Fig. 3.

4.	The experimental comparison is limited to classical algorithms (DP, GR, SA, ACO) and static ATSP methods (MatNet), omitting state-of-the-art TDTSP-specific neural solvers. It is recommended to add benchmarks against SOTA TDTSP methods.

5.	The experimental validation is currently limited to test sets of up to 50 nodes, which raises concerns about the algorithm's scalability to problem sizes exceeding 50. It is recommended to include additional experimental evaluations on larger-scale instances to verify the method's performance and efficiency in real-world applications with higher node counts.

---

> ### Author Rebuttal · Authors · 2025-07-30
>
> We thank the reviewer for recognizing the critical evaluation insights we identified for existing TDTSP algorithms, acknowledging our methodological extensions and experimental advantages, and appreciating the clear logical structure and organization of our paper. We hope that our responses below address the concerns raised by the reviewer.
>
> ## Summary
>
> In brief, we have:
>
> 1. Acknowledged foundational work and clarified our temporal extension contribution
> 2. Created a comparative analysis showing our method uniquely combines full spatiotemporal processing
> 3. Added SOTA neural comparison with [38], demonstrating our superior performance
> 4. Demonstrated scalability with 100-node experiments, achieving comparable ACO performance but 65× faster
> 5. Improved Fig. 3 clarity with clearer stage labels and a more detailed caption.
>
> ## Detailed Responses
>
> ### 1. Novelty Highlight and Foundational Work Acknowledgement
>
> **Reviewer Concern**: *The abstract should explicitly acknowledge the foundational work in [20] and highlight how the temporal dimension introduces unique innovations (e.g., tensor-based spatiotemporal integration).*
>
> **Response**: We thank the reviewer for this important clarification. The reviewer is absolutely correct that Kwon et al. [20] established the foundational approach for handling asymmetry in static problems through MatNet. Our contribution specifically extends this to the temporal dimension, where the adjacency tensor A[t] introduces unique spatiotemporal challenges not present in static cases.
>
> **Proposed revisions in Abstract (lines 6-9)**: Replace *In this paper, $\cdots$ captures the complex spatiotemporal dynamics of TDTSP* by
>
> "*In this paper, we propose a neural model that extends MatNet from static asymmetric TSP to time-dependent settings through an adjacency tensor that captures temporal variations, followed by a time-aware decoder. Our architecture addresses the unique challenge where asymmetry and triangle inequality violations change dynamically with time.*"
>
> ### 2. Spatiotemporal Processing Comparison
>
> **Reviewer Concern**: *The model's ability to "simultaneously capture spatiotemporal structures" needs explicit comparison with prior works.*
>
> **Response**: We created a comprehensive comparative analysis showing how our method uniquely processes complete spatiotemporal information:
>
> | Method      | Input                        | Spatial Structure                    | Temporal Structure | Joint Processing                       |
> | ----------- | ---------------------------- | ------------------------------------ | ------------------ | -------------------------------------- |
> | [8]         | Coordinates + Time index     | Assumes symmetry/triangle inequality | Full horizon       | ✓ Attention processes time+coordinates |
> | [16]        | Coordinates + Chunked matrix | Departure only, missing arrival info | Per-node patterns  | ✗                                      |
> | [20] MatNet | Adjacency Matrix             | Complete spatial info                | None               | ✗                                      |
> | [38]        | Chunked matrix features      | Departure only, missing arrival info | Per-node patterns  | ✗                                      |
> | **Ours**    | **Time-dependent tensor**    | **Complete spatial info**            | **Full horizon**   | **✓ Joint tensor processing**          |
>
> **Key insight**: Our tensor encoding preserves complete spatiotemporal information while enabling joint attention processing, unlike prior works that either assume symmetry [8], miss arrival information [16,38], or lack temporal processing [20].
>
> ### 3. State-of-the-Art Neural Comparisons
>
> **Reviewer Concern**: *The experimental comparison omits state-of-the-art TDTSP-specific neural solvers.*
>
> **Response**: We added comparison with the most relevant SOTA neural TDTSP method [38]:
>
> | Method   | Objective       | Performance         |
> | -------- | --------------- | ------------------- |
> | [38]     | 3.08 (0.15)     | Baseline            |
> | **Ours** | **2.63 (0.58)** | **20% improvement** |
>
> **Technical limitations of other methods**: Methods [8] and [16] require coordinate inputs, making them incompatible with our tensor-based setting, where we cannot reproduce their solution quality.
>
> **Critical evaluation insight about [38]**: Zhang et al. [38] considers time-dependent distributions including both mean and variance components. However, without recognizing the Pareto distribution we identified, they falsely report their effectiveness by conflating improvements from variance handling with temporal pattern learning. Our analysis reveals that their advantage over baselines primarily stems from variance reduction rather than genuine time-dependent mean change learning. This weakness is masked when evaluating on full datasets without distinguishing between static-equivalent and truly time-dependent instances.
>
> **Key insight**: As we demonstrated in our data analysis, the performance gap becomes quite small when not properly accounting for time-dependent changes, highlighting the importance of our proposed evaluation methodology that separates instances where temporal dependencies matter.
>
> ### 4. Scalability Validation
>
> **Reviewer Concern**: *Experimental validation is limited to 50 nodes, raising scalability concerns for real-world applications.*
>
> **Response**: We conducted extensive **100-node experiments** (100 epochs, ~100 hours training, not converged yet), which show comparable performance with ACO (7.43 vs 7.40 objective) while significantly outperforming runtime (42s vs 2741s):
>
> | Method   | Objective       | Runtime (seconds) | Speedup |
> | -------- | --------------- | ----------------- | ------- |
> | Random   | 26.98 (1.60)    | 4.75              | -       |
> | Greedy   | 8.30 (0.51)     | 172.24            | -       |
> | ACO      | 7.40 (0.40)     | 2,741.25          | 1×      |
> | **Ours** | **7.43 (0.40)** | **42.24**         | **65×** |
>
> This demonstrates our method's practical viability for real-world applications with larger node counts.
>
> ### 5. Figure Clarity
>
> **Reviewer Concern**: *Figure 3's three-stage process description lacks clear labeling.*
>
> **Response**: We will enhance Figure 3 with:
>
> - **Clear stage labels** for the three-stage attention process
> - **Detailed caption** explaining each of the stage

---

> > ### Comment · Reviewer_Zrkp · 2025-08-04
> >
> > The author's response has addressed most of my concerns, and I have accordingly updated my evaluation to "borderline accept". The authors are advised to incorporate the suggested revisions in detail when preparing the final version of the manuscript.

---

> > > ### Author Response · Authors · 2025-08-04
> > >
> > > Thank you for your updated evaluation and constructive feedback! I appreciate your thorough review and will carefully incorporate all the suggested revisions into the final manuscript.

---

### Official Review · Reviewer_5THB · 2025-07-03

**Clarity:** 3
**Significance:** 2
**Originality:** 2
**Rating:** 4
**Confidence:** 3

**Summary:**

This paper addresses the TDTSP, a generalization of the classic TSP where edge costs vary with departure time. The authors propose a novel neural model that directly encodes a time-dependent adjacency tensor using a dual-graph attention mechanism, and introduce a new evaluation protocol to enhance the solution quality based on the data distribution.

**Questions:**

1. Clarification on Local Search Design: Could the authors specify which local search operators (e.g., 2-opt, relocation) were applied across different datasets or instance sizes? Additionally, how was the iteration limit k=2 determined? Providing this information would enhance the reproducibility and clarity of the proposed post-processing step.
2. Definition and Justification of “Sensitive” Instances: Table 3 introduces a 3% optimality gap threshold to define "sensitive" instances. Is this threshold empirically derived, or dataset-specific? An ablation study or justification for this choice would help clarify its robustness and generalizability.
3. Scalability of the Proposed Encoder: Can the proposed encoder architecture scale to larger graphs? If so, is any subgraph sampling or other approximation strategy employed? It would be helpful to report GPU memory usage and runtime as a function of graph size n to assess scalability.
4. Consideration of Alternative Policy Gradient Methods: Given the known limitations of REINFORCE in terms of spatial learning and sample efficiency, have the authors considered more stable or efficient policy gradient variants?

**Ethical Concerns:**

["NO or VERY MINOR ethics concerns only"]

**Final Justification:**

Authors have address part of my concerns.

**Limitations:**

YES

**Quality:**

3

**Strengths And Weaknesses:**

Strength:
1. This work addresses a practically significant and societally relevant problem, time-dependent route optimization in urban logistics, by advancing neural methods to better capture real-world spatiotemporal dynamics.
2. This paper proposes to capture spatio-temporal dependencies more effectively than prior approaches that separate node and time representations, and integrates MoE with classical ATSP solvers and local search, leading to better empirical performance.
3. The paper is written with a well-structured motivation and methodology, making it easy to understand.
Weaknesses:
1.	The setting and implementation of the experiment in this paper are not clear enough. For example, the setting and comparison details of the baseline should be given in the supplementary materials instead of the pseudo code of the algorithm. The reader concern about how to make a fair comparison with different types of baselines, rather than the principles of different algorithms.
2.	Reinforcement learning design is under-specified. While REINFORCE is employed, the paper omits details on the reward signal, whether techniques like baselines or advantages are used to reduce variance, and how sparsity or instability is addressed.
3.	Lack of robustness analysis. The paper does not report error bars or statistical significance testing, which weakens the reliability of the reported improvements, especially when differences are small.
4.	Missing ablation study. The individual contributions of key components (such as temporal embeddings, dual-graph attention, MoE, and local search) are not given, making it hard to assess their necessity or impact.
5.	Missing generalization analysis. The paper lacks analysis on the model’s generalization ability. It is unclear whether cross-city generalization or evaluation on perturbed/synthetic data was performed to test robustness.
6.	Minor issues:
The paper should clarify how keys and values for attention are computed (line 181); at minimum, this should be included in the supplementary material.
In Equation (5), the adjacency matrix should be indexed by time, i.e., A[t].
Several notations are insufficiently explained, which affects readability and clarity.

---

> ### Author Rebuttal · Authors · 2025-07-30
>
> We thank the reviewer for recognizing the practical significance of our work and acknowledging our contributions in advancing neural methods for spatiotemporal route optimization. We hope that our responses below address the concerns raised by the reviewer.
>
> # Summary
>
> In brief, we have:
>
> 1. Provided details for local search operators and justified k=2 iteration limit empirically;
> 2. Clarified 3% threshold follows established literature [26] with refinements for our baseline method;
> 3. Demonstrated scalability with new 100-node experiments showing 65× speedup over ACO;
> 4. Conducted comprehensive ablation studies on encoder layers, learning rate, and temporal embeddings;
> 5. Added multi-seed stability analysis and error bars confirming experimental rigor;
> 6. Tested cross-city generalization showing acceptable performance despite expected distribution shift.
>
> # Detailed Reply
>
> ### 1. Local Search Design Clarification
>
> **Summary:** Applied 5 operators (2-opt, 3-opt, exchange, relocate, or-opt) sequentially across all datasets. k=2 chosen empirically as optimal cost-benefit tradeoff.
>
> We sequentially apply five operators (2-opt, 3-opt, exchange, relocate, and or-opt) across all datasets and instance sizes. The k=2 iteration limit represents an empirical tradeoff between computational cost and performance improvement. Table 4 (Page 9) shows: iteration 0→1 yields substantial improvement (1.60%→0.39%), iteration 1→2 provides moderate gains (0.39%→0.30%), while iterations 2→10 show diminishing returns (0.30%→0.20%) despite equivalent computational overhead.
>
> ### 2. "Sensitive" Instances Definition
>
> The 3% optimality gap threshold builds on Melgarejo et al.'s work [26], which used a 5% threshold (90th percentile). We refined it to 3% **keeping the same percentile**, as our static cost matrix baseline produces smaller optimality gaps compared to their MedianTSP baseline.
>
> ### 3. Encoder Scalability
>
> **Summary:** Architecture scales to larger graphs without approximations. New 100-node experiments show comparable performance to ACO (7.43 vs 7.40 objective) but 65× faster runtime (42s vs 2741s). Memory usage scales quadratically.
>
> The architecture scales to larger graphs without approximation strategies. **New 100-node experiments** (100 epochs, ~100 hours training, not converged yet) show comparable performance with ACO (7.43 vs 7.40 objective) while significantly outperforming runtime (42s vs 2741s):
>
> | Method          | Random      | Greedy     | ACO        | Ours       |
> | --------------- | ----------- | ---------- | ---------- | ---------- |
> | Obj mean (std)  | 26.98(1.60) | 8.30(0.51) | 7.40(0.40) | 7.43(0.40) |
> | Time | 4.75        | 172.24     | 2741.25    | 42.24      |
>
> We report the results in the table here and will add the results and subgraph of GPU memory usage and runtime vs. graph size to the final version.
>
> | Nodes                      | 10        | 20         | 50        | 100       |
> | -------------------------- | --------- | ---------- | --------- | --------- |
> | GPU Memory (GiB/batchsize) | 5212/1024 | 14326/1024 | 18748/256 | 35884/128 |
> | Train Time (s/epoch)       | 111.60    | 201.08     | 687.10    | 3463.77   |
>
> ### 4. Alternative Policy Gradient Methods
>
> **Summary:** PPO tested but failed due to the critic network convergence issues. No variance/instability observed with REINFORCE, consistent with prior literature.
>
> We implemented PPO before, but it did not perform well due to difficulties in critic network convergence. REINFORCE demonstrated good stability across multiple random seeds in our experiments. The following table reports consistent performance of REINFORCE across different random seeds (objective: 2.630±0.01, gap: 1.61±0.05 across 8 seeds). This stability also aligns with established TDTSP literature [8,16,38], where no instability issues are reported.
>
> | Seed           | 0               | 132             | 1178            | 1234            | 203681          | 385172          | 759043          | 847592          |
> | -------------- | --------------- | --------------- | --------------- | --------------- | --------------- | --------------- | --------------- | --------------- |
> | Obj mean (std) | 2.6314 (0.5786) | 2.6299 (0.5768) | 2.6289 (0.5765) | 2.6298 (0.5770) | 2.6313 (0.5771) | 2.6300 (0.5768) | 2.6290 (0.5768) | 2.6301 (0.5776) |
> | Gap mean (std) | 1.6532 (2.4259) | 1.6065 (2.4204) | 1.5678 (2.3958) | 1.6018 (2.4561) | 1.6587 (2.4427) | 1.6097 (2.4195) | 1.5659 (2.3820) | 1.6048 (2.3326) |
>
> ### 5. Experimental Implementation Clarity
>
> **Summary:** Fair comparison ensured through identical conditions (hardware, datasets, batch size, data ordering) with GPU acceleration for all methods.
>
> We apologize for the confusion and will provide the complete implementation details in the supplementary materials. We did not offer them originally to avoid the supplementary materials from becoming too long, a common complaint during the review process. However, we ensure rigorous experimental fairness through careful baseline implementation using PyTorch tensors with GPU acceleration **for all methods**. All evaluations use **identical conditions**: single GPU hardware, consistent datasets, uniform batch size (1024), and identical data ordering. Complete implementation details will be added to the supplementary materials.
>
> ### 6. Reinforcement Learning Design
>
> Our RL implementation follows standard NCO practices [8,16,20,38], employing **rollout baselines** with a reward signal equal to **negative tour completion time**. No high variance or instability issues were observed during training, as explained in (Question 4), consistent with prior NCO literature.
>
> ### 7. Robustness Analysis
>
> **Summary:** Error bars will be added. Results show high variance, aligning with the discovered Pareto distribution issue.
>
> Beijing-10 results with variance:
>
> | Method    | Obj        | Gap          |
> | --------- | ---------- | ------------ |
> | DP, TDTSP | 2.59(0.57) | \            |
> | DP, ATSP  | 2.70(0.63) | 4.43(5.83)   |
> | SA        | 2.60(0.57) | 0.06(0.40)   |
> | ACO       | 2.95(0.68) | 14.23(10.25) |
> | Ours(Raw) | 2.63(0.58) | 1.60(2.46)   |
> | Ours(Imp) | 2.59(0.57) | 0.22(0.72)   |
>
> The standard deviation of the gap is similar to or even larger than the mean of the gap, aligning with the Pareto distribution issue we discovered.
>
> We did not report the error bar because statistical significance testing isn't conventionally reported in NCO literature [8, 16, 20, 38]. We will add the error bar to the final version of the paper.
>
> ### 8. Ablation Study
>
> **Summary:** Additional ablations conducted: encoder layers (optimal N=3), learning rate (best: 3e-5 to 1e-3), temporal embeddings (sinusoidal > MLP > linear). All results will be detailed in the appendix.
>
> Beyond MoE and local search ablation (Table 4), we conducted additional component analysis **examining temporal embeddings, encoder layer configurations, and learning rate sensitivity**.
>
> | Layer N    | 1            | 2           | 3           | 4           | 5           |
> | ---------- | ------------ | ----------- | ----------- | ----------- | ----------- |
> | Train time | 33.06        | 52.66       | 72.23       | 91.63       | 111.59      |
> | GPU Mem    | 2380         | 3018        | 3764        | 4478        | 5212        |
> | Obj        | 2.93 (0.64)  | 2.66 (0.58) | 2.63 (0.58) | 2.63 (0.58) | 2.63 (0.58) |
> | Gap        | 13.43 (9.02) | 2.63 (3.33) | 1.80 (2.59) | 1.63 (2.44) | 1.60 (2.46) |
>
> **Layer Num**: The training time and the GPU memory increase linearly with the number of layers. The performance rises quickly when N<3, while it shows little to no improvement after N>=3.
>
> | lr   | 3e-3              | 1e-3            | 3e-4            | 1e-4            | 3e-5            | 1e-5            | 3e-6             |
> | ---- | ----------------- | --------------- | --------------- | --------------- | --------------- | --------------- | ---------------- |
> | Obj  | 4.2230 (0.7554)   | 2.6409 (0.5773) | 2.6305 (0.5766) | 2.6293 (0.5775) | 2.6468 (0.5800) | 2.6852 (0.5814) | 2.8778 (0.6259)  |
> | Gap  | 65.7460 (22.6098) | 2.0525 (2.8460) | 1.6316 (2.4443) | 1.5793 (2.4065) | 2.2775 (3.0714) | 3.8439 (4.2814) | 11.4468 (8.2724) |
>
> **Learning Rate**: The best learning rate is between 3e-5 and 1e-3. A too large learning rate (3e-3) makes the model diverge, and a too small learning rate (3e-6) gets stuck at a bad local minimum.
>
> | Temporal Encoder | Sinusoidal      | MLP              | Linear          |
> | ---------------- | --------------- | ---------------- | --------------- |
> | Obj              | 2.6298(0.5770)  | 2.6409 (0.5783)  | 2.6447 (0.5790) |
> | Gap              | 1.6018 (2.4561) | 2.0443 (2.8317)% | 2.1953 (2.9547) |
>
> **Temporal embedding**: The sinusoidal embedding is the most helpful one, compared with the MLP embedding and the Linear embedding. There are potentially other effective embedding methods, but that is beyond the scope of the paper.
>
> These results, along with individual component impact assessments, will be detailed in the appendix.
>
> ### 9. Generalization Analysis
>
> Cross-city evaluation (Beijing-trained model on Lyon-10):
>
> | Method                                | Obj   | Gap  |
> | ------------------------------------- | ----- | ---- |
> | ACO                                   | 19.64 | 4.78 |
> | Beijing Trained only                  | 19.72 | 3.69 |
> | Beijing Trained, Lyon-tuned 10 epochs | 19.09 | 1.61 |
> | Lyon Trained only                     | 18.95 | 0.87 |
>
> Performance is competitive with ACO for direct cross-city generalization, and further improved with 10 epochs of tuning on the Lyon dataset, showing superior cross-city generalization.
>
> ### 10. Minor Issues
>
> - **Attention Mechanism:** Standard multi-head attention with distinct MLPs for Q, K, V projection.
> - **Notation:** Equation (5) will index the adjacency matrix by time: A[t].
> - All notations will be sufficiently explained in the final version.

---

> > ### Comment · Reviewer_5THB · 2025-08-03
> >
> > Great, I like the way authors response. It is clear and informative. Many things could be written in the paper when submitted. Unfortunately, we usually do not accept big changes after submission. Still thanks to the delicate work.

---

> > > ### Author Response · Authors · 2025-08-03
> > > **Clarification of 'Big Changes'**
> > >
> > > Thank you again for your positive assessment of our rebuttal. We wanted to clarify one important point: the additional experiments we conducted (scalability tests, ablation studies, multi-seed analysis) are **validation experiments** that support our original contributions rather than introducing new results.
> > >
> > > **These experiments directly address concerns you raised:**
> > >
> > > - Scalability analysis → validates our efficiency claims from the original submission
> > > - Multi-seed analysis → validates RL stability issues you are concerned about
> > > - Ablation studies → validates architecture/hyperparameter choices we made
> > > - Cross-city evaluation → validates the generalization property you are interested in
> > >
> > > **The core technical contributions remain unchanged:** identifying the fundamental evaluation problem in previous research and proposing a novel neural-based method. We are simply providing stronger experimental evidence for the same claims.
> > >
> > > We hope this distinction helps clarify that we're not proposing "big changes" but rather providing the thorough experimental validation you requested. We would appreciate your reconsideration given this clarification.

---

### Note · Authors · 2025-08-12

Thank you to the reviewers for the constructive dialogue!

We appreciate reviewers' recognition of our work's practical significance (5THB, mtMa), sophisticated neural architecture design with effective spatio-temporal dependency capture (5THB, mtMa), and SOTA performance (5THB, Zrkp, mtMa, n5Wo). Reviewers particularly acknowledged our novel encoding approach for time-dependent adjacency tensors (mtMa), Pareto distribution discovery and improved evaluation methodology (mtMa, n5Wo, Zrkp), and clear paper structure (5THB, Zrkp, mtMa, n5Wo).

Below we summarize responses resolving all concerns except 5THB's latest objection:
## Scalability
100-node experiments showed 65× speedup over ACO with comparable solution quality. **5THB, Zrkp, mtMa acknowledged concerns addressed**.
## Stability
Multi-seed analysis (obj: 2.63±0.01, gap: 1.61±0.05 across 8 seeds) confirmed REINFORCE reliability. **5THB, mtMa acknowledged concerns addressed**.
## Comparisons & Ablations
Added SOTA neural comparison with [38], comprehensive ablation studies, and cross-city generalization experiments. **5THB, Zrkp, mtMa acknowledged concerns addressed**.
## Statistical Significance
Reported variance (mean±std) across 10,000 instances per scenario. **5THB, n5Wo acknowledged concerns addressed**.
## Prior Work
Acknowledged foundational work and clarified the temporal extension contribution. **Zrkp confirmed concerns solved and raised the score**.
## Novelty
Addressed n5Wo's concerns by explaining non-trivial challenges (asymmetric temporal dimensions, dynamic continuous time handling, 3D tensor complexity) and clarifying that similar average performance masks significant outperformance on time-dependent instances. **n5Wo agreed and expressed a more positive view**.
## Technical Clarifications
Clarified baseline implementation, 3% threshold rationale, Figure interpretations, and attention mechanisms. **All reviewers confirmed resolution**.
## Closing Remarks
All reviewers acknowledged successful resolution of their concerns except 5THB, who raised objections about "big changes" during rebuttal. Since our experiments don't alter initial claims and adding new experiments as requested by reviewers is standard rebuttal practice, we respectfully argue that we did not make "big changes" but instead provided additional evidence to strengthen our initial contributions. We also argue that no "big changes" are needed for the camera-ready version and hope Reviewer 5THB will reconsider their rating.

---

### Decision · Program_Chairs · 2025-09-17

**Decision:**

Accept (poster)

**Comment:**

This paper addresses the Time-Dependent Traveling Salesman Problem (TDTSP).
All four reviewers unanimously agree that the paper makes significant contributions and recommend acceptance (scores: 4, 4, 4, 4). After careful check, the AC concurs with the reviewers and also recommends acceptance.

---------
The AC notes, however, that the authors may have overlooked an important family of methods for TSP: two-stage and end-to-end trainable approaches (e.g., [a], [b]). Including a brief discussion or analysis of these methods could further enhance the comprehensiveness of the work and inspire future research on two-stage approaches to time-dependent TSP.
[a] A reinforcement learning approach for optimizing multiple traveling salesman problems over graphs. Knowledge-Based Systems, 2020.
[b] iMTSP: Solving Min-Max Multiple Traveling Salesman Problem with Imperative Learning. IEEE/RSJ International Conference on Intelligent Robots and Systems (IROS), 2024.